# Live streaming mode selection strategy under the background of virtual anchor supplementation

**Shizhen Bai[1], Xiujin Gu[1], Na Xu[2], Jinjin Zheng[1], Wenya Wu[1]\*, Man Jiang[1], Ning Xue[1]**

**1** School of Management, Harbin University of Commerce, Harbin, China, **2** College of Business Administration, Shandong Technology and Business University, Yantai, China

\* wuwenya2021@163.com

## Abstract

In the context of virtual anchors as a supplement to human anchors in live streaming e-commerce. In this paper, for the first time, virtual anchors are included in the live streaming mode selection strategy, combined with the characteristics of the live streaming anchor and the cost of live streaming, constructed a model of the four live streaming modes, and used the Stackelberg game method to study. The results show that: (1) live streaming prices are positively correlated with cross price elasticity coefficient, market share of live streaming channels and consumer sensitivity to live streaming e-commerce. (2) When the cross price elasticity coefficient or market share of live streaming channels is small, the influence of influencer anchors can attract more consumers to purchase products, and the influencer anchor live streaming mode is the optimal choice; When the cross price elasticity coefficient or market share of live streaming channels is large, the merchant combined with virtual anchors live streaming mode is the optimal choice due to the low cost of live streaming for merchant staff anchors, as well as the advantage of continuous live streaming for virtual anchors. (3) When the consumer sensitivity to live streaming e-commerce is small, due to the low cost of live streaming for merchant staff anchors, the merchant live streaming mode is the optimal choice; When the consumer sensitivity to live streaming e-commerce is large, based on the influence of influencer anchors and the advantage of continuous live streaming for virtual anchors, the influencer anchor combined with virtual anchors live streaming mode is the optimal choice.

## 1. Introduction

Live streaming is becoming increasingly popular worldwide [1]. In daily life, both businesses and consumers can sell and purchase products on platforms such as Instagram Live Shopping, Twitter's e-commerce module, Taobao Live, Facebook Live, and Amazon Live [2]. With the development of live streaming e-commerce, new live streaming modes are constantly emerging. Businesses can choose suitable live streaming modes based on factors such as live streaming anchor type, live streaming anchor characteristics, and consumer preferences, such as merchant live streaming mode, influencer anchor live streaming mode, etc. [3–7]. With the development of artificial intelligence technology, using virtual anchors as a supplement

**Data availability statement:** All relevant data are within the manuscript.

**Funding:** This work was supported by Social Science Foundation Program of Heilongjiang province (Grant/Award Number 22GLD356), the funders had no role in study design, data collection and analysis, decision to publish, or preparation of the manuscript.

**Competing interests:** The authors have declared that no competing interests exist.

to human anchors has become a new hybrid live streaming mode. For example, well-known beauty brands such as Carslan, L' Oreal, and Perfect Diary choose merchant staff anchors or influencer anchors such as Viya and Li Jiaqi for live streaming sales. After midnight, they also use virtual anchors such as "Big Eye Kaka", "Ou Xiaomi", and "Stella" to replace human anchors and continue selling products.

The mixed live streaming mode of human anchors and virtual anchors can use virtual anchors to fill the time gap when human anchors cannot live streaming sales. However, the construction and application of virtual anchors require a certain cost investment. At present, the low-end version of virtual anchors on the market is priced at tens of thousands of yuan. Merchants can achieve "7 * 24" hour live streaming sales of virtual anchors by purchasing anchor software and operation and maintenance services. The high-end version of virtual anchors is mostly personalized customization, with different levels of pricing based on the difficulty of customization, with an average price of around 200000 yuan. Due to the poor live streaming interaction ability of virtual anchors, most live streaming rooms choose to use a combination of "human anchors and virtual anchors" for live streaming sales, manufacturers need to pay for the construction and application costs of virtual anchors in addition to the cost of human anchors. Therefore, based on cost and profit considerations, whether manufacturers choose traditional live streaming modes such as merchant live streaming mode and influencer anchor live streaming mode, or choose to invest in and apply virtual anchors on the basis of the traditional live streaming mode to obtain additional income, is a question worth studying.

Based on this, in the context of virtual anchors as a supplement to human anchors in live streaming e-commerce. In this paper, the first time to include virtual anchors in the live streaming mode selection strategy, combined with the characteristics of the live streaming anchor and the cost of live streaming, four live streaming modes are constructed: merchant live streaming mode, influencer anchor live streaming mode, merchant combined with virtual anchors live streaming mode, and influencer anchor combined with virtual anchors live streaming mode. Explore the optimization problem of manufacturer's live streaming mode selection decision. The problems to be solved are as follows:

(1) What are the optimal live streaming channel prices and manufacturers' optimal profits under different live streaming modes?

(2) What impact do parameters such as cross price elasticity coefficient, market share of live streaming channels and consumer sensitivity to live streaming e-commerce have on live streaming channel prices?

(3) What is the manufacturer's live streaming mode selection strategy considering the impact of parameters such as cross price elasticity coefficient, market share of live streaming channels and consumer sensitivity to live streaming e-commerce?

The rest of the paper is organized as follows: section 2, reviews the related literature. Section 3, performs problem description and constructs mathematical models. Section 4, solves and analyzes the equilibrium solution. Section 5, analyzes the impact of parameters and discusses the limitations and future research. Section 6, summarizes the main conclusions of this paper.

## 2. Literature review

The literature related to the research in this paper can be categorized into three areas: live streaming mode selection, virtual anchors, and the characteristics of the live streaming anchor.

## 2.1. Live streaming mode selection

In terms of live streaming mode selection. Du et al. investigated the manufacturer's anchor type selection (celebrity anchor or ordinary anchor) and limited sales strategy (unlimited quantity or limited quantity) [4]. Jin et al. combined the key features of fresh agricultural products and live streaming to establish two live streaming e-commerce modes: merchant live streaming selling and influencer anchor live streaming selling model and explore the influence of parameters such as the probability of harvest of agricultural products on the selection of live streaming mode [8]. Zhang et al. studied two live streaming modes, merchant live streaming selling and influencer anchor live streaming selling, by combining the anchor's live streaming e-commerce level and other variables, and the results showed that the commission rate and the basic remuneration of the influencer anchor had a certain influence on the selection of the manufacturer's live streaming mode [9]. Yang et al. based on three common sales modes in live commerce such as e-commerce platform mode, money trans-fer mode and live streaming platform mode, considered the consumer's return behavior, and used a game theory approach to study the selection of live streaming e-commerce modes [10]. Pan et al. take the anchor's live streaming e-commerce level, consumer's preference, and consumer's purchase cost of live streaming channel as the influencing factors of live streaming e-commerce, and find that adding live streaming channel can only improve profitability if the anchor's selling power is high enough [5]. Ji et al. study the channel selection problem and price discounting strategy of suppliers in the live streaming environment [11]. Fan et al. used the Stackelberg game approach to study the impact of live streaming commercial spillovers on price decisions as well as anchor service efforts [12]. Ma and Yang et al. studied different live streaming e-commerce strategies of manufacturers by con-structing a live streaming e-commerce model for an e-commerce supply chain by combining vertical differences in products [13]. Gong et al. investigated the live streaming strategies of online retailers under multi-channel sales and found that product standardization and product quality affect the profit of e-retailers [14]. Xie et al. investigated the optimal live streaming service strategies of retailers by using a game-theoretic approach and considering freeriding behaviors [15]. Cui et al. constructed a live streaming e-commerce supply chain by combining the features of live streaming e-commerce in a model [16]. Lu and Chen investigated how online sellers can adjust their live streaming approach based on signaling theory to reduce the perceived matching uncertainty of product-centric and social-centric consumers to online sellers and increase their trust [17]. Hao and Yang investigated the resale and agency of live streaming environments in terms of sales and corresponding pricing strategies [18]. Wongkitrungrueng et al. studied live streaming e-commerce methods and strategies to acquire and retain customers [19]. Jiang et al. considered the effects of brand awareness and anchors on consumers' willingness to purchase live streaming and explored the optimal strategic combination of brand and anchor [20]. Xin et al. use the Stackelberg method to study the choice of three e-commerce live sales modes: brand self-live streaming, influencer-led live streaming mixture, and influencer-led special live streaming [21]. Different from previous studies, we synthesize the characteristics of anchors and the cost of live streaming in the context of introducing virtual anchors to study the live streaming pricing and mode selection issues for manufacturers.

## 2.2. Virtual anchors

In terms of virtual anchors. A virtual anchor is an avatar powered by artificial intelligence technology as an anchor on a live streaming platform, with a stable live streaming capability that is not limited by location or time [22]. Scholars have mostly used empirical methods to study the impact of virtual anchors on consumer behavior. Gao et al. explored the impact of

virtual anchors versus human anchors on consumers' purchase intentions, and found that the demand for novelty had a significant moderating effect on consumers' purchase intentions [6]. Um et al. found that compared to consumers with a low degree of novelty, those with a high degree of novelty were more likely to accept new technology and have positive attitudes toward virtual anchors [23]. Xu and Ruan investigated the relative effectiveness of virtual anchors versus human anchors in attracting consumers with different levels of social overload based on the PAD model, and found that virtual anchors were more appealing to consumers with higher levels of social overload [24]. Youn and Jin explored the different relationship types of AI chatbots (e.g., assistant and friend) with brand personality (e.g., competence and sincerity) [25]. Zhang et al. investigated the moderating role of AI-driven virtual anchors in consumer engagement and humor responses [26]. Song et al. found that effective interaction with chatbots affects consumers' behavioral intentions [27]. Current scholars confirmed the existence of a certain influence relationship between virtual anchors on consumer behavior during live streaming, and in this paper, the impact of virtual anchors is expressed in terms of their continuous live streaming level and applied to the construction of a live streaming model.

## 2.3. The characteristics of the live streaming anchor

In terms of the characteristics of the live streaming anchor. Existing literature has mostly utilized empirical methods to analyze the impact of live streaming anchors on consumer purchasing intentions based on the characteristics and performances (e.g., trustworthiness, attractiveness, professional knowledge, and popularity) of different types of live streaming anchors [28–30]. Drawing on the theory of quasi-social interactions and the flow theory, Liao et al. examined the impact of the anchor's interaction orientation on consumer immersion and quasi-social interactions, as well as the effect on consumers' purchase intention [7]. Martinez Lopez et al. found that third-party live streaming anchors (especially key opinion leaders such as Jiaqi Li and Yonghao Luo) were more able to enhance consumers' confidence and trust in recommended products [31]. Lei et al. constructed a theoretical model of the influencer anchor of personality on consumers' purchase intention, finding that influencer anchor personality affects consumers' purchase intention [32]. Roy and Naidoo investigated that the anthropomorphic conversational style of an AI chatbot affects consumers' purchase intention [33]. Crolic et al. explored the effects of anthropomorphism in AI on consumer satisfaction, business evaluation, and purchase intention [34]. Current scholars' research found that credibility, attractiveness, professional knowledge and other the characteristics of the live streaming anchor will affect consumers' purchase intention, this paper is based on attractiveness and other the characteristics of the live streaming anchor, to construct the product demand function of the live streaming channel.

Different from previous studies, this paper, in the context of virtual anchors as a supplement to human anchors in live streaming e-commerce, considers the introduction of virtual anchors from merchant live streaming mode, influencer anchor live streaming mode, respectively, and constructs four different live streaming modes, which enriches the live streaming mode model. Secondly, for the first time, virtual anchors are included in the live streaming mode selection decisions, and the optimal live streaming channel prices and manufacturers' optimal profits are studied by synthesizing live streaming anchor characteristics and live streaming costs. Finally, based on the impact of parameters such as cross price elasticity coefficient, market share of live streaming channels and consumer sensitivity to live streaming e-commerce on manufacturers' profits, it provides a reference for manufacturer's live streaming mode selection decisions.

## 3. Methodology

### 3.1. Description of the problem

In this paper, we consider a live streaming e-commerce supply chain consisting of a manufacturer, an e-retailer, and a live streaming anchor, where the manufacturer resells its products through the e-retailer and establishes a live streaming channel to sell its products. The live streaming modes in the live streaming channel are classified into four types, namely, merchant live streaming mode, influencer anchor live streaming mode, merchant combined with virtual anchors live streaming mode, and influencer anchor combined with virtual anchors live streaming mode, as shown in Fig 1.

In the merchant live streaming mode, the product demand of the live streaming channel is affected by the levels of live sales for merchant staff $u_1$ . The manufacturer invests some of its employees in live streaming anchor training to improve the level of live sales. Referring to the service effort cost function, the training cost function for the level of live sales of merchant staff can be expressed as follows: $C(u_1) = \frac{1}{2}\theta u_1^2$ , $\theta(0 < \theta < 1)$ which is the training cost coefficient for the level of live sales of merchant staff [19].

In the influencer anchor live streaming mode, the product demand of the live streaming channel is affected by the levels of live sales for influencer anchors $u_2$ , the influence level of influencer anchors $h$. The manufacturer pays live commissions and basic honorarium to the net red anchor, the influencer anchor's commission ratio is $\gamma(0 < \gamma < 1)$.

In the merchant combined with virtual anchors live streaming mode, and influencer anchor combined with virtual anchors live streaming mode, the level of investment in continuous live streaming for virtual anchors $g$, will be through the exposure effect of the virtual anchor continuous live streaming and a certain amount of live interaction ability to influence the consumer's willingness to buy. The cost function for the level of investment in continuous live streaming for virtual anchors can be expressed as follows: $C(g) = \frac{1}{2}\lambda g^2$ , $\lambda(0 < \lambda < 1)$ which is the cost coefficients for the level of investment in continuous live streaming for virtual anchors.

Virtual anchors are a new type of live streaming anchor, and consumers' trust in virtual anchors can enhance their willingness to purchase [22,35]. In addition, consumers also have trust issues with the live streaming sales performance of ordinary live streaming anchors, influencer anchors, and other live streaming anchors [28]. Therefore, for ease of analysis, this article does not separately consider consumers' trust in virtual anchors as an influencing factor, but uniformly regards consumers' trust and sensitivity to the characteristics of live streaming anchors (trustworthiness, live streaming interaction ability, etc.) as consumers' sensitivity to live streaming e-commerce, and uses $\varphi(0 < \varphi < 1)$ to represent it [5,28].

In the context of virtual anchors as a supplement to human anchors in live streaming e-commerce, when manufacturers choose to expand their sales channels through live streaming sales, there are four modes: merchant live streaming mode, influencer anchor live streaming mode, merchant combined with virtual anchors live streaming mode, and influencer anchor combined with virtual anchors live streaming mode. In the merchant live streaming mode and the merchant combined with virtual anchors live streaming mode, manufacturers play a dominant role in the game with e-tailers based on their strong financial strength and long-term product development [36–38]. Therefore, when manufacturers establish live streaming channels to sell products, they will prioritize determining the live streaming channel price, and e-tailers will determine the e-tailing channel price based on the manufacturer's live streaming pricing; In the influencer anchor live streaming mode and the influencer

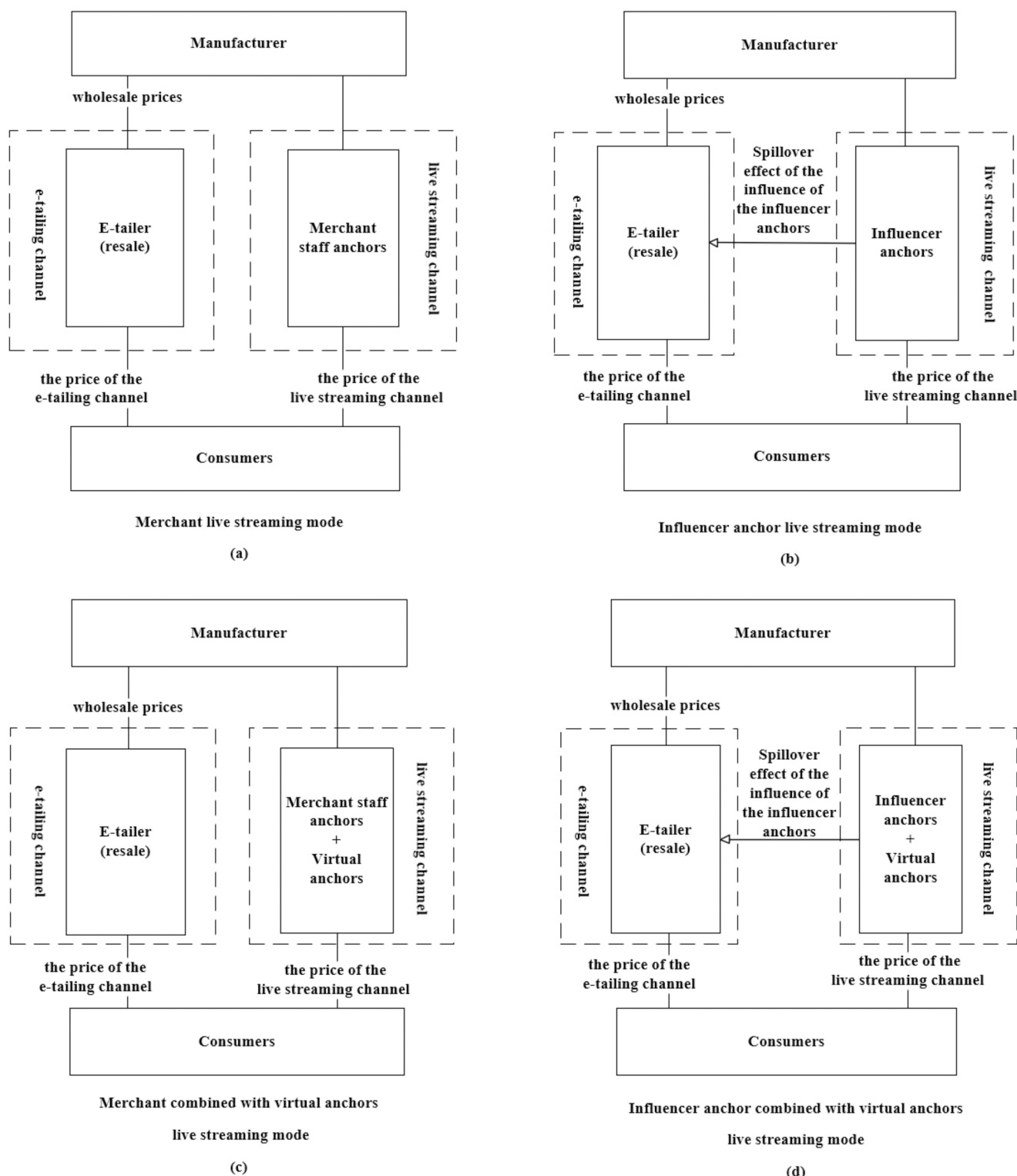

**Fig 1. Dual channel supply chain structures for four different manufacturer live streaming modes.**

anchor combined with virtual anchors live streaming mode, manufacturers hire influencer anchors for live streaming sales, such as well-known influencer anchors with a large number of fans such as Li Jiaqi, Viya, Dong Yuhui, etc. Due to their influence, they have certain pricing power in live streaming sales [10,39]. Therefore, when manufacturers establish live streaming channels to sell products, influencer anchors will first determine the live streaming channel price, and e-tailers will determine the e-tailing channel price based on the live streaming pricing of influencer anchors. The sequence of the game is specifically shown in Fig 2.

## 3.2. Model parameters

The relevant parameters and meanings used in the model of this paper are shown in Table 1.

Where $i \in \{1,2,3,4\}$, respectively denotes merchant live streaming mode, influencer anchor live streaming mode, merchant combined with virtual anchors live streaming mode, and influencer anchor combined with virtual anchors live streaming mode; and $j \in \{e,l\}$, respectively denotes the e-tailing channel and the live streaming channel; and $n \in \{E,M,I\}$, respectively denotes the e-tailer, the manufacturer, and the influencer anchor.

## 3.3. Basic assumptions

To facilitate the analysis of the problem and without compromising the generality of the conclusions, the following modeling assumptions are made in this paper:

(1) In this paper, $k(0 < k < 1)$ denotes the market share of the live streaming channel, and $1 - k$ denotes the market share of the e-tailing channel. $r(0 < r < 0.5)$ denotes the cross price elasticity coefficient between online sales channels, and refers to the sensitivity of a product's supply and demand in a marketing channel to price changes of other related substitutes [18].

(2) For the convenience of the study, this paper assumes that the manufacturer's wholesale price is $w$, which is determined by the long-term contract signed between the manufacturer and the e-retailer, is an exogenous variable of the model.

(3) Combined with reality, it is assumed that the levels of live sales for influencer anchors is the best, the levels of live sales for merchant staff is the second best, i.e., $0 < u_1 < u_2 < 1$.

(4) The cost price of the product produced by the manufacturer is not taken into account, and the commission rate of the live streaming platform is not taken into account.

## 3.4. Demand function

The demand functions for the e-tailing channel and the live streaming channel in the merchant live streaming mode are:

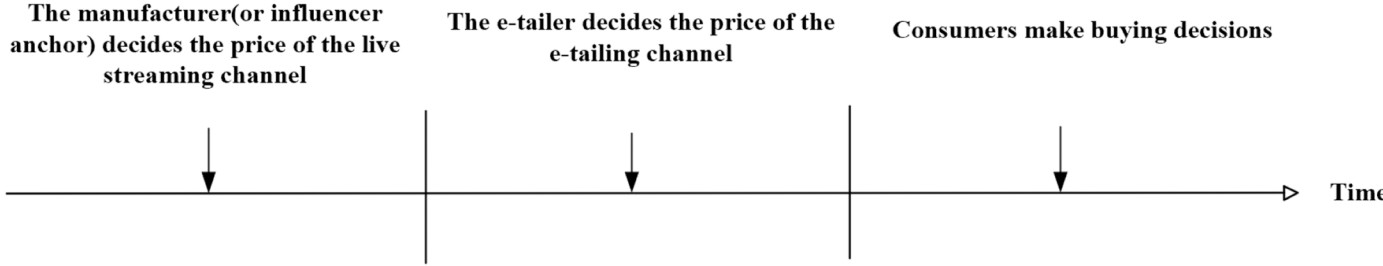

**Fig 2. Sequence of the game.**

**Table 1. Summary of the symbols.**

| Symbols | Descriptions |
|---|---|
| $p_{ij}$ | Product price of channel $j$ in model $i$, decision variables |
| $w$ | Wholesale prices of manufacturer, exogenous variables |
| $a$ | Market demand for the product |
| $k$ | Market share of the live streaming channel |
| $r$ | Cross price elasticity coefficient between online sales channels |
| $u_1$ | Levels of live sales for merchant staff anchors |
| $u_2$ | Levels of live sales for influencer anchors |
| $g$ | Level of investment in continuous live streaming for virtual anchors |
| $h$ | Influence level of influencer anchors |
| $\varphi$ | Consumer sensitivity to live streaming e-commerce |
| $\eta$ | Spillover effect of the influence of the influencer anchors |
| $\sigma$ | Exposure effect of continuous live streaming by virtual anchors |
| $\theta$ | Training cost factor for live sales level of merchant staff anchors |
| $\varepsilon$ | Training cost factor for live sales level of influencer anchors |
| $\lambda$ | Cost coefficients for the level of investment in continuous live streaming for virtual anchors |
| $\gamma$ | Live streaming commission rates for the influencer anchors |
| $\delta$ | Basic remuneration for the influencer anchors |
| $D_{ij}$ | Demand for products of channel $j$ in model $i$ |
| $\pi_{in}$ | Profit of member $n$ in mode $i$ |

$$D_{1e} = (1-k)a - p_{1e} + rp_{1l} \tag{1}$$

$$D_{1l} = ka - p_{1l} + rp_{1e} + \varphi\mu_1 \tag{2}$$

where consumer sensitivity to live streaming e-commerce $\varphi(0 < \varphi < 1)$ denotes the degree of consumer preference for live sales shopping.

The demand functions for the e-tailing channel and the live streaming channel in the influencer anchor live streaming mode are:

$$D_{2e} = (1-k)a - p_{2e} + rp_{2l} + \eta h \tag{3}$$

$$D_{2l} = ka - p_{2l} + rp_{2e} + \varphi(\mu_2 + h) \tag{4}$$

In this case, the spillover effect of the influence of the influencer anchors $\eta(0 < \eta < 0.5)$ indicates that the personal influence based on the influencer anchor can stimulate consumers to purchase this product in the e-tailing channel [10].

The demand functions for the e-tailing channel and the live streaming channel in the merchant combined with virtual anchors live streaming mode are:

$$D_{3e} = (1-k)a - p_{3e} + rp_{3l} \tag{5}$$

$$D_{3l} = ka - p_{3l} + rp_{3e} + \varphi(\mu_1 + g) + \sigma g \tag{6}$$

Among them, the exposure effect of continuous live streaming by virtual anchors $\sigma(0 < \sigma < 1)$ indicates the extent of the application of virtual anchors for continuous live streaming, which will indirectly increase the exposure of the live room in the live streaming platform, which in turn affects the consumer's willingness to buy.

The demand functions for the e-tailing channel and the live-streaming channel in the influencer anchor combined with virtual anchors live streaming mode are:

$$D_{4e} = (1-k)a - p_{4r} + rp_{4l} + \eta h \tag{7}$$

$$D_{4l} = ka - p_{4l} + rp_{4e} + \varphi(\mu_2 + h + g) + \sigma g \tag{8}$$

## 4. Results

### 4.1. Merchant live streaming mode

The profit function of the e-tailer and the manufacturer are as follows:

$$\pi_{1E} = (p_{1e} - w)D_{1e} = (p_{1e} - w)(a - ak - p_{1e} + rp_{1l}) \tag{9}$$

$$
\begin{aligned}
\pi_{1M} &= p_{1l}D_{1l} + wD_{1e} - \frac{\theta u_1^2}{2} \\
&= p_{1l}(ak - p_{1l} + rp_{1e} + \varphi u_1) + w(a - ak - p_{1e} + rp_{1l}) - \frac{\theta u_1^2}{2}
\end{aligned}
\tag{10}
$$

The first-order derivative of the e-tailer's profit function $\pi_{1E}$ concerning the e-tailing channel price $p_{1e}$ is:

$$\frac{\partial \pi_{1E}}{\partial p_{1e}} = w - 2p_{1e} + rp_{1l} - ak + a \tag{11}$$

From Equation (11), we can see that $\frac{\partial \pi_{1E}^2}{\partial p_{1e}^2} = -2 < 0$; the e-tailer's profit function is a concave function of the price of the e-tailing channel, so there is a maximum value of the e-tailer's profit function. To Equation (11) for the first-order derivative condition on $p_{1e}$ can be obtained under the e-tailer's profit maximization of the optimal price solution, so that $\frac{\partial \pi_{1E}}{\partial p_{1e}} = 0$, can be obtained e-tailing channel price equilibrium solution:

$$p_{1e} = \frac{rp_{1l} + w - ak + a}{2} \tag{12}$$

The first order derivative of the manufacturer's profit function $\pi_{1M}$ with respect to the live streaming channel price $p_{1l}$ is:

$$\frac{\partial \pi_{1M}}{\partial p_{1l}} = -p_{1l} + ak + \varphi u_1 + \frac{2rw - 2p_{1l}(1-r^2) + ar(1-k)}{2} \tag{13}$$

From [Equation (13)](), we can see that $\frac{\partial \pi_{1M}^2}{\partial p_{1l}^2} = r^2 - 2 < 0, \left(0 < r < 0.5\right)$; The manufacturer's profit function $\pi_{1M}$ is a concave function with respect to the price of the live streaming channel $p_{1l}$, hence there exists a maximum value of the manufacturer's profit function $\pi_{1M}$.

Substituting [Eq. (11)]() into [Eq. (12)]() and taking the first-order derivative of the live streaming channel price, the live streaming channel price equilibrium solution can be obtained:

$$p_{1l}^* = \left(ak + \varphi u_1\right)/\left(2 - r^2\right) + \left(ar - ark + 2rw\right)/2\left(2 - r^2\right) \tag{14}$$

Substituting [Eq. (14)]() into [Eq. (11)](), the e-tailing channel price equilibrium is solved as:

$$\begin{aligned}p_{1e}^* = \left(w + a - ak\right)/2 + \left(ak + \varphi u_1\right)r/2\left(2 - r^2\right) + \\ \left(2w + a - ak\right)r^2/4\left(2 - r^2\right)\end{aligned} \tag{15}$$

In price equilibrium, the e-tailer's profit and the manufacturer's profit are:

$$\pi_{1E}^* = \frac{\left(4a - 4w - 4ak - ar^2 + 4r^2w + 2akr + 2r\varphi u_1 + akr^2\right)^2}{16\left(2 - r^2\right)^2} \tag{16}$$

$$\begin{aligned}\pi_{1M}^* = \left(a^2k^2 - 2a^2k + a^2 + 4\theta u_1^2 + 8w^2\right)r^2/8\left(2 - r^2\right) + \\ \left(a^2k - a^2k^2 + a\varphi u_1 + 2akw + 2w\varphi u_1 - ak\varphi u_1\right)r/2\left(2 - r^2\right) + \\ \left(a^2k^2 + 2ak\varphi u_1 - 2akw + 2aw + \varphi^2 u_1^2 - 2\theta u_1^2 - 2w^2\right)/2\left(2 - r^2\right)\end{aligned} \tag{17}$$

## 4.2. Influencer anchor live streaming mode

The profit function of the e-tailer, the manufacturer, and the influencer anchor are as follows:

$$\pi_{2E} = \left(p_{2e} - w\right)D_{2e} = \left(p_{2e} - w\right)\left(a - ak - p_{2e} + rp_{2l} + \eta h\right) \tag{18}$$

$$\begin{aligned}\pi_{2M} &= \left(1 - \gamma\right)p_{2l}D_{2l} + wD_{2e} - \delta \\ &= \left(1 - \gamma\right)p_{2l}\left(ak - p_{2l} + rp_{2e} + \varphi u_2 + \varphi h\right) + \\ &\quad w\left(a - ak - p_{2e} + rp_{2l} + \eta h\right) - \delta\end{aligned} \tag{19}$$

$$\begin{aligned}\pi_{2I} &= \gamma p_{2l}D_{2l} + \delta - \frac{\varepsilon u_2^2}{2} \\ &= \gamma p_{2l}\left(ak - p_{2l} + rp_{2e} + \varphi u_2 + \varphi h\right) + \delta - \frac{\varepsilon u_2^2}{2}\end{aligned} \tag{20}$$

The first-order derivative of the e-tailer's profit function $\pi_{2E}$ concerning the e-tailing channel price $p_{2e}$ is:

$$\frac{\partial \pi_{2E}}{\partial p_{2e}} = w - 2p_{2e} + rp_{2l} + \eta h + a - ak \tag{21}$$

From [Equation (21)](), we can see that $\frac{\partial \pi_{2E}^2}{\partial p_{2e}^2} = -2 < 0$; the e-tailer's profit function is a concave function of the price of the e-tailing channel, so there is a maximum value of the e-tailer's profit function. To [Equation (21)]() for the first-order derivative condition on $p_{2e}$ can

be obtained under the e-tailer's profit maximization of the optimal price solution, so that $\frac{\partial \pi_{2E}}{\partial p_{2e}} = 0$, can be obtained e-tailing channel price equilibrium solution:

$$p_{2e} = \frac{w + \eta h + rp_{2l} + a - ak}{2} \tag{22}$$

The first order derivative of the profit function of influencer anchor $\pi_{2I}$ with respect to the live streaming channel price $p_{2l}$ is:

$$\frac{\partial \pi_{2I}}{\partial p_{2l}} = \gamma\left(\varphi h + \varphi u_2 - p_{2l} + ak\right) + \frac{\gamma\left(rw + r\eta h + 2r^2 p_{2l} + ar - ark - 2p_{2l}\right)}{2} \tag{23}$$

From Equation (23), we can see that $\frac{\partial \pi_{2I}^2}{\partial p_{2l}^2} = \gamma\left(r^2 - 2\right) < 0, \left(0 < r < 0.5\right)$; The profit function of influencer anchor $\pi_{2I}$ is a concave function with respect to the price of the live streaming channel $p_{2l}$, hence there exists a maximum value of the profit function of influencer anchor $\pi_{2I}$.

Substituting Eq. (21) into Eq. (22) and taking the first-order derivative of the live streaming channel price, the live streaming channel price equilibrium solution can be obtained:

$$p_{2l}^* = \left(ak + \varphi u_2 + \varphi h\right) / \left(2 - r^2\right) + r\left(a + w + \eta h - ak\right) / 2\left(2 - r^2\right) \tag{24}$$

Substituting Eq. (24) into Eq. (21), the e-tailing channel price equilibrium is solved as:

$$p_{2e}^* = \left(w + \eta h + a - ak\right) / 2 + \left(ak + \varphi u_2 + \varphi h\right) r / 2\left(2 - r^2\right) + \left(w + \eta h + a - ak\right) r^2 / 4\left(2 - r^2\right) \tag{25}$$

In price equilibrium, the profit function of the e-tailer, the manufacturer, and the influencer anchor are:

$$\pi_{2E}^* = \frac{\left(4a - 4w + 4\eta h - 4ak - ar^2 + 3r^2 w + 2akr + 2\varphi hr + 2\varphi ru_2 - \eta hr^2 + akr^2\right)^2}{16\left(2 - r^2\right)^2} \tag{26}$$

$$\pi_{2M}^* = \begin{bmatrix} \left(1-\gamma\right)\left(a^2 - 2a^2 k + \eta^2 h^2 + a^2 k^2 - 2\eta ahk + 2\eta ah\right) + 8\delta - \\ \gamma w^2 + 7w^2 - 2\gamma aw - 2\gamma \eta hw + 2\gamma akw \end{bmatrix} r^2 / 8\left(2 - r^2\right) +$$
$$\begin{bmatrix} \left(1-\gamma\right)\begin{pmatrix} a^2 k - a^2 k^2 + a\varphi h + a\varphi u_2 + \eta \varphi h^2 - a\varphi ku_2 + \eta ahk + \\ \eta \varphi hu_2 - a\varphi hk \end{pmatrix} + \\ \left(2-\gamma\right)\left(akw + \varphi hw + \varphi u_2 w\right) \end{bmatrix} r / 2\left(2 - r^2\right) + \tag{27}$$
$$\begin{bmatrix} \left(1-\gamma\right)\left(a^2 k^2 + \varphi^2 h^2 + \varphi^2 u_2^2 + 2a\varphi hk + 2a\varphi ku_2 + 2\varphi^2 hu_2\right) + \\ 2aw + 2\eta hw - 2akw - 2w^2 - 4\delta \end{bmatrix} / 2\left(2 - r^2\right)^2$$

$$\pi_{2I}^* = \begin{bmatrix} \gamma\begin{pmatrix} \eta^2 h^2 - 2\eta ahk + 2\eta ah + 2\eta hw + a^2 k^2 - 2a^2 k + \\ a^2 - 2akw + 2aw + w^2 \end{pmatrix} - 8\delta + 4\varepsilon u_2^2 \end{bmatrix} r^2 / 8\left(2 - r^2\right) +$$
$$\begin{bmatrix} \gamma\begin{pmatrix} a^2 k - a^2 k^2 + a\varphi h + a\varphi u_2 + akw + \varphi hw + \varphi u_2 w + \eta \varphi h^2 + \\ \eta ahk + \eta \varphi hu_2 - a\varphi ku_2 - a\varphi hk \end{pmatrix} \end{bmatrix} r / 2\left(2 - r^2\right) + \tag{28}$$
$$\begin{bmatrix} \gamma\left(a^2 k^2 + \varphi^2 h^2 + 2\varphi^2 u_2^2 + 2a\varphi hk + 2a\varphi ku_2 + 2\varphi^2 hu_2\right) + \\ 4\delta - 2\varepsilon u_2^2 \end{bmatrix} / 2\left(2 - r^2\right)$$

### 4.3. Merchant combined with virtual anchors live streaming mode

The profit function of the e-tailer and the manufacturer are as follows:

$$\pi_{3E} = \left(p_{3e} - w\right)D_{3e} = \left(p_{3e} - w\right)\left(a - ak - p_{3e} + rp_{3l}\right) \tag{29}$$

$$
\begin{aligned}
\pi_{3M} &= p_{3l}D_{3l} + wD_{3e} - \frac{\theta u_1^2}{2} - \frac{\lambda g^2}{2} \\
&= p_{3l}\left(ak - p_{3l} + rp_{3e} + \varphi u_1 + \varphi g + \sigma g\right) + \\
&\quad w\left(a - ak - p_{3e} + rp_{3l}\right) - \frac{\theta u_1^2}{2} - \frac{\lambda g^2}{2}
\end{aligned}
\tag{30}
$$

The first-order derivative of the e-tailer's profit function $\pi_{3E}$ concerning the e-tailing channel price $p_{3e}$ is:

$$\frac{\partial \pi_{3E}}{\partial p_{3e}} = w - 2p_{3e} + rp_{3l} + a - ak \tag{31}$$

From Equation (31), we can see that $\dfrac{\partial \pi_{3E}^2}{\partial p_{3e}^2} = -2 < 0$; the e-tailer's profit function is a concave function of the price of the e-tailing channel, so there is a maximum value of the e-tailer's profit function. To Equation (31) for the first-order derivative condition on $p_{3e}$ can be obtained under the e-tailer's profit maximization of the optimal price solution, so that $\dfrac{\partial \pi_{3E}}{\partial p_{3e}} = 0$, can be obtained e-tailing channel price equilibrium solution:

$$p_{3e} = \frac{w + rp_{3l} + a - ak}{2} \tag{32}$$

The first order derivative of the manufacturer's profit function $\pi_{3M}$ with respect to the live streaming channel price $p_{3l}$ is:

$$\frac{\partial \pi_{3M}}{\partial p_{3l}} = ak + r^2 p_{3l} - 3p_{3l} + \sigma g + \varphi u_1 + \varphi g + rw + \frac{ar}{2} - \frac{akr}{2} \tag{33}$$

From Equation (33), we can see that $\dfrac{\partial \pi_{3M}^2}{\partial p_{3l}^2} = r^2 - 2 < 0, \left(0 < r < 0.5\right)$; The manufacturer's profit function $\pi_{3M}$ is a concave function with respect to the price of the live streaming channel $p_{3l}$, hence there exists a maximum value of the manufacturer's profit function $\pi_{3M}$.

Substituting Eq. (31) into Eq. (32) and taking the first-order derivative of the live streaming channel price, the live streaming channel price equilibrium solution can be obtained:

$$p_{3l}^* = \left(ak + \varphi g + \varphi u_1 + \sigma g\right)/\left(2 - r^2\right) + \left(2w + a - ak\right)r / 2\left(2 - r^2\right) \tag{34}$$

Substituting Eq. (34) into Eq. (21), the e-tailing channel price equilibrium is solved as:

$$
\begin{aligned}
p_{3e}^* &= \left(w + a - ak\right)/2 + \left(\varphi g + ak + \sigma g + \varphi u_1\right)r / 2\left(2 - r^2\right) + \\
&\quad \left(2w + a - ak\right)r^2 / 4\left(2 - r^2\right)
\end{aligned}
\tag{35}
$$

In price equilibrium, the e-tailer's profit and the manufacturer's profit are:

$$\pi_{3E}^* = \frac{(4a - 4w - 4ak - ar^2 + 4r^2w + 2\varphi gr + 2akr + 2\sigma gr + 2\varphi ru_1 + akr^2)^2}{16(2 - r^2)^2} \tag{36}$$

$$\pi_{3M}^* = \left(a^2k^2 - 2a^2k + a^2 + 4\theta u_1^2 + 4\lambda g^2 + 8w^2\right)r^2 / 8\left(2 - r^2\right) +$$
$$\left(\begin{array}{l} a^2k - a^2k^2 + a\varphi g + a\sigma g + a\varphi u_1 + 2\varphi gw + 2\sigma gw + \\ 2akw + 2w\varphi u_1 - ak\varphi u_1 - agk\varphi - agk\sigma \end{array}\right) r / 2\left(2 - r^2\right) + \tag{37}$$
$$\left(\begin{array}{l} a^2k^2 + 2agk\varphi + 2agk\sigma + 2ak\varphi u_1 - 2akw + 2aw + \varphi^2 g^2 + \varphi^2 u_1^2 + \\ \sigma^2 g^2 - 2\theta u_1^2 - 2w^2 + 2\varphi\sigma g^2 - 2\lambda g^2 + 2\varphi^2 gu_1 + 2g\varphi\sigma u_1 \end{array}\right) / 2\left(2 - r^2\right)$$

## 4.4. Influencer anchor combined with virtual anchors live streaming mode

The profit function of the e-tailer, the manufacturer, and the influencer anchor are as follows:

$$\pi_{4E} = \left(p_{4e} - w\right)D_{4e} = \left(p_{4e} - w\right)\left(a - ak - p_{4e} + rp_{4l} + \eta h\right) \tag{38}$$

$$\pi_{4M} = (1 - \gamma)p_{4l}D_{4l} + wD_{4e} - \delta - \frac{\lambda g^2}{2}$$
$$= (1 - \gamma)p_{4l}\left(ak - p_{4l} + rp_{4e} + \varphi u_2 + \varphi h + \varphi g + \sigma g\right) + \tag{39}$$
$$w\left(a - ak - p_{4e} + rp_{4l} + \eta h\right) - \delta - \frac{\lambda g^2}{2}$$

$$\pi_{4I} = \gamma p_{4l}D_{4l} + \delta - \frac{\varepsilon u_2^2}{2}$$
$$= \gamma p_{4l}\left(ak - p_{4l} + rp_{4e} + \varphi u_2 + \varphi h + \varphi g + \sigma g\right) + \delta - \frac{\varepsilon u_2^2}{2} \tag{40}$$

The first-order derivative of the e-tailer's profit function $\pi_{4E}$ concerning the e-tailing channel price $p_{4e}$ is:

$$\frac{\partial \pi_{4E}}{\partial p_{4e}} = w - 2p_{4e} + rp_{4l} + \eta h + a - ak \tag{41}$$

From Equation (41), we can see that $\frac{\partial \pi_{4E}^2}{\partial p_{4e}^2} = -2 < 0$, ; the e-tailer's profit function is a concave function of the price of the e-tailing channel, so there is a maximum value of the e-tailer's profit function. To Equation (38) for the first-order derivative condition on $p_{4e}$ can be obtained under the e-tailer's profit maximization of the optimal price solution, so that $\frac{\partial \pi_{4E}}{\partial p_{4e}} = 0$, can be obtained e-tailing channel price equilibrium solution:

$$p_{4e} = \frac{w + \eta h + rp_{2l} + a - ak}{2} \tag{42}$$

The first order derivative of the profit function of influencer anchor $\pi_{4I}$ with respect to the live streaming channel price $p_{4l}$ is:

$$\frac{\partial \pi_{4I}}{\partial p_{4l}} = \gamma\left(\varphi h + \varphi u_2 + \varphi g + \sigma g - p_{2l} + ak\right) + \frac{\gamma\left(rw + r\eta h + 2r^2 p_{2l} + ar - ark - 2p_{2l}\right)}{2} \tag{43}$$

From Equation (43), we can see that $\dfrac{\partial \pi_{4I}^2}{\partial p_{4l}^2} = \gamma\left(r^2 - 2\right) < 0, \left(0 < r < 0.5\right)$; The profit function of influencer anchor $\pi_{4I}$ is a concave function with respect to the price of the live streaming channel $p_{4l}$, hence there exists a maximum value of the profit function of influencer anchor $\pi_{4I}$.

Substituting Eq. (41) into Eq. (42) and taking the first-order derivative of the live streaming channel price, the live streaming channel price equilibrium solution can be obtained:

$$p_{4l}^* = \left(ak + \varphi u_2 + \varphi h + \varphi g + \sigma g\right) / \left(2 - r^2\right) + r\left(a + w + \eta h - ak\right) / 2\left(2 - r^2\right) \tag{44}$$

Substituting Eq. (44) into Eq. (41), the e-tailing channel price equilibrium is solved as:

$$p_{4m}^* = \left(w + \eta h + a - ak\right) / 2 + \left(ak + \varphi u_2 + \varphi h + \varphi g + \sigma g\right) r / 2\left(2 - r^2\right) + \\ \left(w + \eta h + a - ak\right) r^2 / 4\left(2 - r^2\right) \tag{45}$$

In price equilibrium, the profit function of the e-tailer, the manufacturer, and the influencer anchor are:

$$\pi_{4E}^* = \frac{\left(4a - 4w + 4\eta h - 4ak - ar^2 + 3r^2 w + 2\varphi rg + 2\sigma gr + 2akr + 2\varphi hr + 2\varphi ru_2 - \eta hr^2 + akr^2\right)^2}{16\left(2 - r^2\right)^2} \tag{46}$$

$$\pi_{4M}^* = \begin{bmatrix} \left(1-\gamma\right)\left(a^2 - 2a^2 k + \eta^2 h^2 + a^2 k^2 - 2\eta ahk + 2\eta ah\right) + 8\delta - \\ \gamma w^2 + 7w^2 - 2\gamma aw - 2\gamma\eta hw + 2\gamma akw + 4\lambda g^2 \end{bmatrix} r^2 / 8\left(2 - r^2\right) + \\ \begin{bmatrix} \left(1-\gamma\right)\begin{pmatrix} a^2 k - a^2 k^2 + a\varphi g + a\varphi h + a\varphi u_2 + a\sigma g + \eta\varphi h^2 + \eta g\varphi h + \\ \eta ahk + \eta gh\sigma + \eta\varphi hu_2 - ag\varphi k - a\varphi hk - agk\sigma - a\varphi ku_2 \end{pmatrix} + \\ \left(2-\gamma\right)\left(akw + \varphi hw + \varphi u_2 w + g\varphi w + g\sigma w\right) \end{bmatrix} r / 2\left(2 - r^2\right) + \tag{47} \\ \begin{bmatrix} \left(1-\gamma\right)\begin{pmatrix} g^2\varphi^2 + a^2 k^2 + \varphi^2 h^2 + \sigma^2 g^2 + \varphi^2 u_2^2 + 2g\varphi^2 h + 2g^2\varphi\sigma + \\ 2g\varphi^2 u_2 + 2a\varphi hk + 2a\varphi ku_2 + 2\varphi^2 hu_2 + 2g\varphi\sigma u_2 + 2ag\varphi k + \\ 2ag\sigma k + 2g\varphi h\sigma \end{pmatrix} + \\ 2aw + 2\eta hw - 2akw - 2w^2 - 2\lambda g^2 - 4\delta \end{bmatrix} / 2\left(2 - r^2\right)^2$$

$$\pi_{2I}^* = \begin{bmatrix} \gamma\begin{pmatrix} \eta^2 h^2 - 2\eta ahk + 2\eta ah + 2\eta hw + a^2 k^2 - 2a^2 k + \\ a^2 - 2akw + 2aw + w^2 \end{pmatrix} - 8\delta + 4\varepsilon u_2^2 \end{bmatrix} r^2 / 8\left(2 - r^2\right) + \\ \begin{bmatrix} \gamma\begin{pmatrix} a^2 k - a^2 k^2 + ag\varphi + a\varphi h + ag\sigma + a\varphi u_2 + g\varphi w + akw + \\ \varphi hw + g\sigma w + \varphi u_2 w + \eta\varphi h^2 + \eta g\varphi h + \eta ahk + \eta gh\sigma + \\ \eta\varphi hu_2 - ag\varphi k - agk\sigma - a\varphi ku_2 - a\varphi hk \end{pmatrix} \end{bmatrix} r / 2\left(2 - r^2\right) + \tag{48} \\ \begin{bmatrix} \gamma\begin{pmatrix} a^2 k^2 + 2ag\varphi k + 2agk\sigma + 2a\varphi hk + 2a\varphi ku_2 + g^2\varphi^2 + \\ g^2\varphi\sigma + g^2\sigma^2 + 2g\varphi^2 h + 2g\varphi^2 u_2 + 2g\varphi h\sigma + 2g\varphi\sigma u_2 + \\ \varphi^2 h^2 + \varphi^2 u_2^2 + 2\varphi^2 hu_2 \end{pmatrix} + \\ 4\delta - 2\varepsilon u_2^2 \end{bmatrix} / 2\left(2 - r^2\right)$$

## 4.5. Equilibrium solution analysis

**Proposition 1** Under four different live streaming modes, there is a positive relationship between the live streaming channel price and the cross price elasticity coefficient, i.e.,

$$\frac{\partial p_{1l}^*}{\partial r} > 0, \frac{\partial p_{2l}^*}{\partial r} > 0, \frac{\partial p_{3l}^*}{\partial r} > 0, \frac{\partial p_{4l}^*}{\partial r} > 0 \ .$$

**Proof:**

$$\frac{\partial p_{1l}^*}{\partial r} = \left\{ \left[a(1-k)+2w\right]r^2 + 4(ak+\varphi u_1)r + 2\left[a(1-k)+2w\right] \right\} / 2\left(2-r^2\right)^2,$$

$$\frac{\partial p_{2l}^*}{\partial r} = \left\{ \left[a(1-k)+w+\eta h\right]r^2 + 4(ak+\varphi u_2+\varphi h)r + 2\left[a(1-k)+w+\eta h\right] \right\} / 2\left(2-r^2\right)^2,$$

$$\frac{\partial p_{3l}^*}{\partial r} = \left\{ \left[a(1-k)+2w\right]r^2 + 4(ak+\varphi u_2+\varphi g+\sigma g)r + 2\left[a(1-k)+2w\right] \right\} / 2\left(2-r^2\right)^2,$$

$$\frac{\partial p_{4l}^*}{\partial r} = \left\{ \left[a(1-k)+w+\eta h\right]r^2 + 4(ak+\varphi u_2+\varphi h+\varphi g+\sigma g)r + 2\left[a(1-k)+w+\eta h\right] \right\} / 2\left(2-r^2\right)^2 \ .$$

Because $0 < r < 0.5$, $0 < k < 1$, it is obtained $2-r^2 > 0$, $1-k > 0$, therefore

$$\frac{\partial p_{1l}^*}{\partial r} > 0, \frac{\partial p_{2l}^*}{\partial r} > 0, \frac{\partial p_{3l}^*}{\partial r} > 0, \frac{\partial p_{4l}^*}{\partial r} > 0 \ .$$

**Proposition 1** shows that as the cross price elasticity coefficient increases, the manufacturer's optimal live streaming price also increases. The larger the cross price elasticity coefficient, the higher the substitutability of products from live streaming channels to products from online retail channels. Live streaming products have a certain competitive advantage, and the probability of consumers purchasing live streaming products will increase. Manufacturers will increase the live streaming prices of their products in order to increase live streaming sales revenue.

**Proposition 2** Under four different live streaming modes, there is a positive relationship between the live streaming channel price and the market share of the live streaming channel,

i.e., $\frac{\partial p_{1l}^*}{\partial k} > 0, \frac{\partial p_{2l}^*}{\partial k} > 0, \frac{\partial p_{3l}^*}{\partial k} > 0, \frac{\partial p_{4l}^*}{\partial k} > 0 \ .$

**Proof:**

$\frac{\partial p_{1l}^*}{\partial k} = \frac{a(1-r)}{2(2-r^2)}, \frac{\partial p_{2l}^*}{\partial k} = \frac{a(1-r)}{2(2-r^2)}, \frac{\partial p_{3l}^*}{\partial k} = \frac{a(1-r)}{2(2-r^2)}, \frac{\partial p_{4l}^*}{\partial k} = \frac{a(1-r)}{2(2-r^2)}$ . Because $0 < r < 0.5$, it is

obtained $2-r^2 > 0$, $1-r > 0$, therefore $\frac{\partial p_{1l}^*}{\partial k} > 0, \frac{\partial p_{2l}^*}{\partial k} > 0, \frac{\partial p_{3l}^*}{\partial k} > 0, \frac{\partial p_{4l}^*}{\partial k} > 0 \ .$

**Proposition 2** shows that as the market share of live streaming channels increases, the optimal live streaming price for manufacturers also increases. This indicates that when consumers are more inclined to purchase products from live streaming channels, as the proportion of

consumer groups in live streaming channels increases, manufacturers can gain more potential consumers. At this time, manufacturers will choose to increase product sales prices in order to increase profits.

**Proposition 3** Under four different live streaming modes, there is a positive relationship between the live streaming channel price and the consumer sensitivity to live streaming e-commerce, i.e., $\frac{\partial p_{1l}^*}{\partial \varphi} > 0, \frac{\partial p_{2l}^*}{\partial \varphi} > 0, \frac{\partial p_{3l}^*}{\partial \varphi} > 0, \frac{\partial p_{4l}^*}{\partial \varphi} > 0$ .

**Proof:**

$\frac{\partial p_{1l}^*}{\partial \varphi} = \frac{u_1}{2 - r^2}, \frac{\partial p_{2l}^*}{\partial \varphi} = \frac{h + u_2}{2 - r^2}, \frac{\partial p_{3l}^*}{\partial \varphi} = \frac{g + u_1}{2 - r^2}, \frac{\partial p_{4l}^*}{\partial \varphi} = \frac{g + h + u_2}{2 - r^2}$ . Because $0 < r < 0.5$ , it is

obtained $2 - r^2 > 0$ . Therefore $\frac{\partial p_{1m}^*}{\partial \varphi} > 0$ , $\frac{\partial p_{2m}^*}{\partial \varphi} > 0$ , $\frac{\partial p_{3m}^*}{\partial \varphi} > 0$ , $\frac{\partial p_{4m}^*}{\partial \varphi} > 0$ .

**Proposition 3** shows that as the consumer sensitivity to live streaming e-commerce increases, manufacturers' optimal live streaming prices also increase accordingly. This indicates that as consumers become more sensitive to live streaming sales and shopping methods, they are more susceptible to the influence of live streaming sales performance such as product introductions and interactive live streaming, which in turn generates shopping demand. At this time, manufacturers will choose to increase the live streaming prices of their products in order to increase profit income.

**Proposition 4** Under four different live streaming modes, there is a positive relationship between manufacturers' profits and the consumer sensitivity to live streaming e-commerce, i.e., $\frac{\partial \pi_{1M}^*}{\partial \varphi} > 0, \frac{\partial \pi_{2M}^*}{\partial \varphi} > 0, \frac{\partial \pi_{3M}^*}{\partial \varphi} > 0, \frac{\partial \pi_{4M}^*}{\partial \varphi} > 0$ .

**Proof:**

$$\frac{\partial \pi_{1M}^*}{\partial \varphi} = u_1 \left(2ak + 2\varphi u_1 + 2rw + ar - akr\right)/2\left(2 - r^2\right) > 0, (0 < k < 1, 0 < r < 0.5);$$

$$\frac{\partial \pi_{2M}^*}{\partial \varphi} = \left(h + u_2\right)\left(1 - \gamma\right)\left(2ak + 2\varphi h + 2\varphi u_2 + \eta hr + ar - akr\right)/2\left(2 - r^2\right)$$
$$+ \left(h + u_2\right)\left(2 - \gamma\right)rw/2\left(2 - r^2\right) > 0, (0 < y < 1, 0 < k < 1, 0 < r < 0.5);$$

$$\frac{\partial \pi_{3M}^*}{\partial \varphi} = \left(g + u_1\right)\left(2\varphi g + 2ak + 2\sigma + 2\varphi u_1 + 2rw + ar - akr\right)/2\left(2 - r^2\right) > 0,$$
$$(0 < k < 1, 0 < r < 0.5);$$

$$\frac{\partial \pi_{4M}^*}{\partial \varphi} = \left(g + h + u_2\right)\left(1 - \gamma\right)\left(2\varphi g + 2ak + 2\varphi h + 2g\sigma + 2\varphi u_2 + \eta hr + ar - akr\right)/2\left(2 - r^2\right) +$$
$$\left(g + h + u_2\right)\left(2 - \gamma\right)rw/2\left(2 - r^2\right) > 0, (0 < y < 1, 0 < k < 1, 0 < r < 0.5).$$

**Proposition 4** shows that as consumers become more familiar and sensitive to live streaming e-commerce, they are more likely to have the desire to purchase products due to the influence of live streaming product explanations and interactive activities, which increases the sales volume of products through live streaming channels and ultimately improves the profits of manufacturers.

## 5. Discussion

Numerical simulation is carried out with the help of MATLAB to take the profit income as the basis for the selection of the manufacturer's live mode, and to explore the strategy of the manufacturer's live mode selection by comparing the size of the profit under different modes. In order not to lose the generality, we refer to the value range of the initial parameters in the numerical analysis part of the study of Zhang et al, Yang et al [9,10]. And the real-life situation to set the values of the relevant parameters in this study as follows: $a = 500$, $w = 20$, $c = 10$, $u_1 = 7$, $u_2 = 8$, $g = 6$, $h = 6$, $\gamma = 0.2$, $\delta = 1000$, $\eta = 0.3$, $\theta = 200$, $\lambda = 30$, $\varepsilon = 500$, $\sigma = 0.2$.

### 5.1. The impact of the cross price elasticity coefficient on live streaming mode selection

Assuming that the range of the cross price elasticity coefficient $r$ is $r \in (0, 0.5)$, when the values of other parameters are fixed, compare the manufacturer's profit function under four live streaming modes, as shown in Fig 3.

As shown in Fig 3, when the cross price elasticity coefficient changes, the profit changes of the four live streaming modes are analyzed and expressed as follows. Firstly, as the cross price elasticity coefficient increases, the profit of the merchant combined with virtual anchors

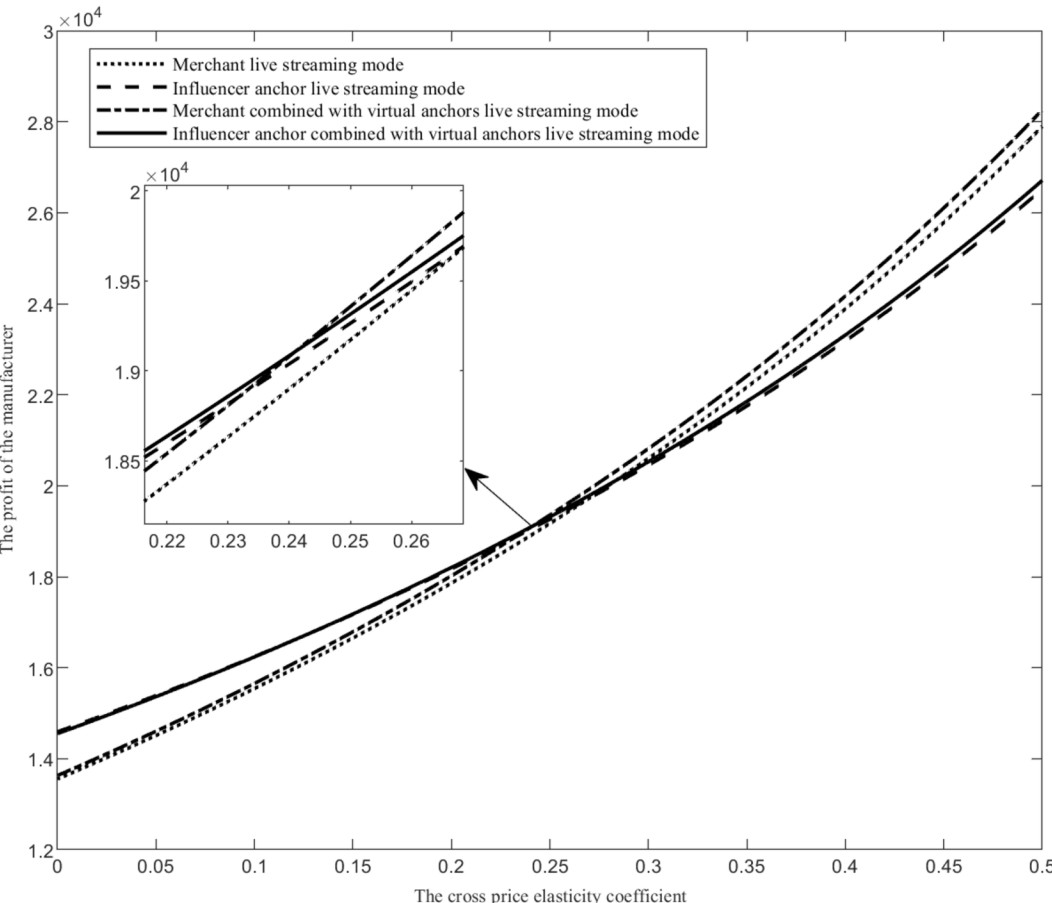

**Fig 3. The impact of the cross price elasticity coefficient on live streaming mode selection.**

live streaming mode (the influencer anchor combined with virtual anchors live streaming mode) is higher than that of the merchant live streaming mode (the influencer anchor live streaming mode). When the cross price elasticity coefficient increases, the change in demand for products in online retail channels is more easily affected by the price of live streaming products. The higher the substitutability of products in live streaming channels to products in online retail channels, the more manufacturers can supplement live streaming sales with virtual anchors, increase live streaming exposure, and make more consumers aware of live streaming products, thereby increasing live streaming sales revenue. Secondly, when the cross price elasticity coefficient is low, the influencer anchor live streaming mode is the optimal choice. When the cross price elasticity coefficient is low, the substitutability of products from live streaming channels to products from online retail channels is small, and products from live streaming channels do not have product advantages. At this time, merchant staff anchors do not have enough attractiveness to consumers, while influencer anchors can attract more consumer attention and purchase products due to their personal influence. Hiring influencer anchors for live sales can bring more income. Thirdly, when the cross price elasticity coefficient is high, the merchant combined with virtual anchors live streaming mode is the optimal choice. When the cross price elasticity coefficient is large, the substitutability of products from live streaming channels to products from online retail channels is higher, and products from live streaming channels have a certain competitive advantage. Compared with the higher commission rate of influencer anchors, the live streaming cost of merchant staff anchors is lower, and manufacturers gain more profits. Therefore, when the cross price elasticity coefficient is large, merchant combined with virtual anchors live streaming mode is the optimal choice.

## 5.2. The impact of the market share of the live streaming channel on live streaming mode selection

Assuming that the value range of the market share of the live streaming channel $k$ is $k \in (0,1)$, when the values of other parameters are fixed, compare the manufacturer's profit function under four live streaming modes, as shown in Fig 4.

As shown in Fig 4, when the market share of live streaming channels changes, the profit changes of the four live streaming modes are analyzed and expressed as follows. Firstly, as the market share of live streaming channels increases, the profit of the merchant combined with virtual anchors live streaming mode (the influencer anchor combined with virtual anchors live streaming mode) gradually exceeds that of the merchant live streaming mode (the influencer anchor live streaming mode). With the increase of market share in live streaming channels, the online consumer group on live streaming channels is gradually growing. More and more consumers will choose to purchase products from live streaming channels. At this time, introducing virtual anchors can increase the exposure of products on live streaming platforms, attract more consumers to understand and purchase products, and improve the profits of manufacturers. Secondly, when the market share of live streaming channels is relatively small, the influencer anchor live streaming mode is the optimal choice. When the market share of live streaming channels is small, consumers have a relatively small proportion in the live streaming channel. Due to the influence of influencer anchors in the consumer group, hiring influencer anchors for live streaming sales is more likely to attract consumers to purchase products and obtain more sales revenue. Thirdly, when the market share of live streaming channels is relatively large, the merchant combined with virtual anchors live streaming mode is the optimal choice. With the increase of market share in live streaming channels, the proportion of consumers in live streaming channels will increase. Based on the live sales level of merchant staff anchors and the advantage of continuous live streaming of virtual anchors,

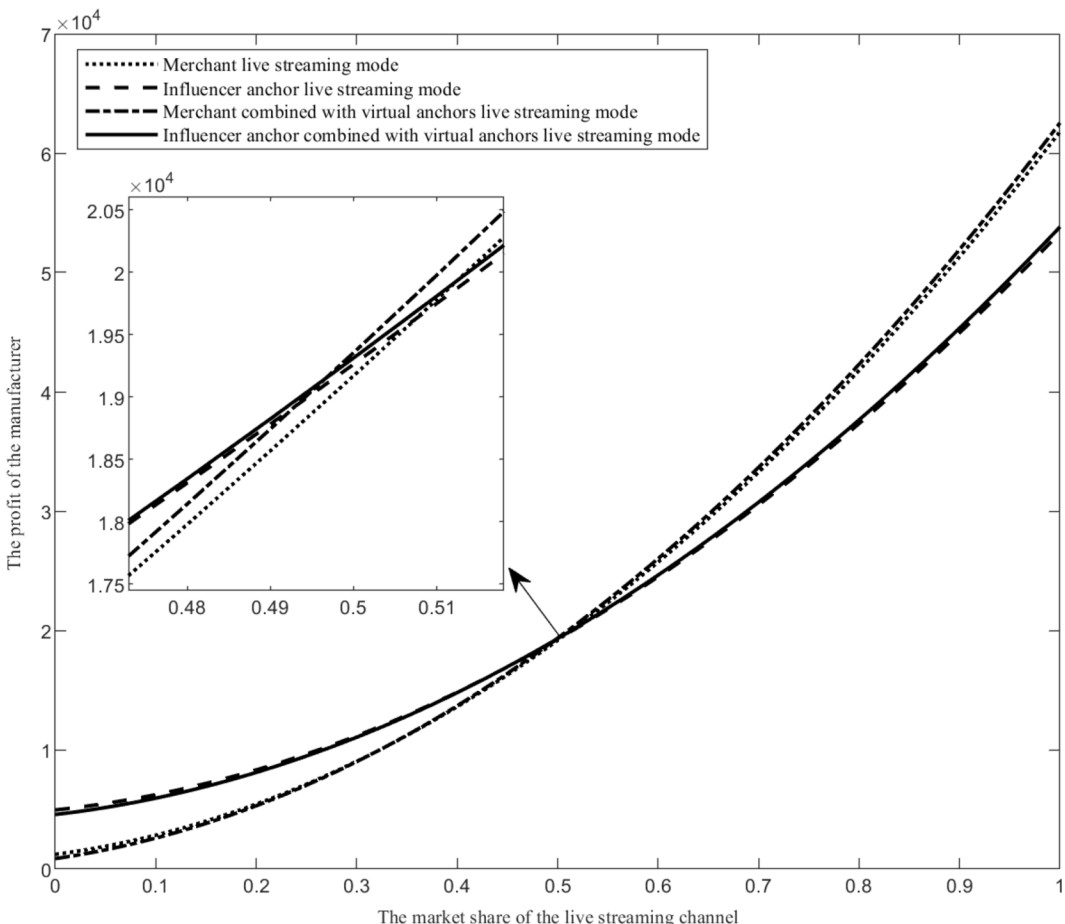

**Fig 4. The impact of the market share of the live streaming channel on live streaming mode selection.**

more consumers can be attracted to purchase products, and the live streaming cost of merchant staff anchors is lower, resulting in more profits for manufacturers.

### 5.3. The impact of the consumer sensitivity to live streaming e-commerce on live streaming mode selection

Assuming that the range of values of the consumer sensitivity to live streaming e-commerce $\varphi$ is $\varphi \in (0,1)$, when the value of other parameters is fixed, compare the manufacturer's profit function under four live streaming modes, as shown in Fig 5.

As shown in Fig 5, when the consumer sensitivity to live streaming e-commerce changes, the profit changes of the four live streaming modes are analyzed and described as follows. Firstly, as the consumer sensitivity to live streaming e-commerce increases, the profit of the merchant combined with virtual anchors live streaming mode (influencer anchor combined with virtual anchors live streaming mode) is higher than that of the merchant live streaming mode (influencer anchor live streaming mode). Introducing virtual anchors can enable manufacturers to obtain more profits. With the consumer sensitivity to live streaming e-commerce increases, live streaming sales can be accepted by a large number of consumers. Under the influence of live streaming sales effect such as product introduction and live streaming interaction, consumers are more likely to have shopping desire. Introducing virtual anchors can

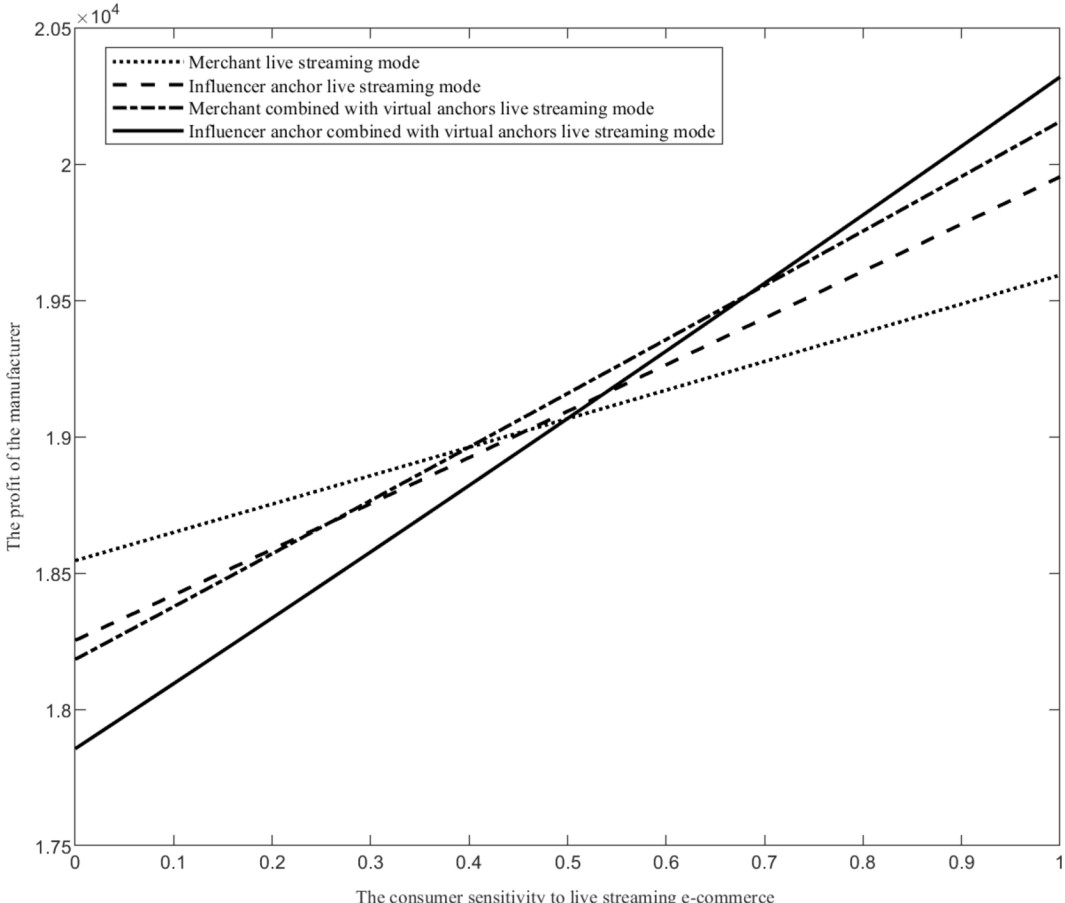

**Fig 5. The impact of the consumer sensitivity to live streaming e-commerce on live streaming mode selection.**

leverage the novelty and uninterrupted live streaming of virtual anchors to enable manufacturers to obtain more profits. Secondly, when the consumer sensitivity to live streaming e-commerce is small, the merchant live streaming mode is the optimal live streaming mode. When consumers are less sensitive to live streaming sales, the probability of purchasing products under the influence of live streaming sales effect is also small. At this time, the introduction of virtual anchors has little effect and requires a certain investment cost. Compared with influencer anchors, the live streaming cost of merchant staff anchors is lower. Therefore, the merchant live streaming mode is the optimal choice. Thirdly, when the consumer sensitivity to live streaming e-commerce is large, the influencer anchor combined with virtual anchors live streaming mode is the optimal live streaming mode. When consumers are more sensitive to live streaming sales, they are more likely to generate shopping needs under the influence of live streaming sales effects such as product introductions by live streaming anchor. With the personal influence of influencer anchors and the advantage of continuous live streaming by virtual anchors, more consumers' attention and purchases can be attracted through live streaming channels, allowing manufacturers to gain more profits.

## 5.4. Discussion

Xin et al. discuss brand self-live streaming, influencer-led live streaming mixture, and influencer-led special live streaming in terms of the anchor's live streaming service level and

consumers' sensitivity to live streaming, and find that influencer-led special live streaming is superior to brand self-live streaming as consumers' sensitivity to live streaming increases [21]. In this paper, in the context of virtual anchors as a supplement to human anchors in live streaming e-commerce, the study finds that consumers' sensitivity to live sales sensitivity on the choice of live streaming mode, there are different findings. Firstly, when the consumer sensitivity to live streaming e-commerce is large, the profit of the merchant combined with virtual anchors live streaming mode gradually exceeds that of influencer anchor live streaming mode. This is because as consumers become more sensitive to live streaming sales, they are more likely to generate shopping desire under the influence of live streaming sales performance such as product introductions and live streaming interactions. At this time, using virtual anchors to supplement live streaming sales of human anchors can leverage the exposure effect of continuous live streaming by virtual anchors. Compared with influencer anchors, the live streaming cost of merchant staff anchors is lower. Therefore, the merchant combined with virtual anchors live streaming mode will gradually be better than the influencer anchor live streaming mode. Secondly, when the consumer sensitivity to live streaming e-commerce is large, the influencer anchor combined with virtual anchors live streaming mode is the optimal live streaming mode, relying on the personal influence of influencer anchors and the continuous exposure of virtual anchors, will attract more consumers' attention and purchase of products, becoming the best choice for manufacturers.

### 5.5. Limitations and future work

This paper has some limitations that can be expanded in future research. Firstly, consumers are influenced by live anchors to produce impulse buying behavior, which can easily lead to the return of live products, after which the issue of live mode selection can be studied by combining the expected regret and return rate of consumers; secondly, live channels can be broadly divided into content platforms represented by TikTok and e-commerce platforms represented by Taobao, and the issue of live mode selection can be studied by combining the operation mechanism and preferential subsidy mechanism of different live platforms; furthermore, e-tailing channels generally exist in two ways, resale by e-tailers and sales by e-commerce platforms, and the choice of resale and agency sales can be studied later.

## 6. Conclusions

In this paper, in the context of virtual anchors to supplement the human anchor live with goods, combined with the anchor characteristics and live operating costs, constructed a model of the four live modes, solved and analyzed the manufacturer's optimal live pricing, to explore the cross price elasticity coefficient, the market share of the live streaming channel and the consumer sensitivity to live streaming e-commerce on the manufacturer's profit, the analysis results provide certain reference value for manufacturers' pricing and live streaming mode selection decisions.

The research results show that:

(1) When the cross price elasticity coefficient, market share of live streaming channels, or consumer sensitivity to live streaming sales increase, the optimal live streaming price for manufacturers also increases. Based on consumers' trust and recognition of live streaming sales, consumers are more inclined to purchase products from live streaming channels, the demand for products in live streaming channels will increase, and manufacturers will choose to increase live streaming price in order to obtain more profits.

(2) When the cross price elasticity coefficient increases, the higher the substitutability of products in live streaming channels to products in online retail channels, the

manufacturers can supplement live streaming sales with virtual anchors, increase live streaming exposure, and make more consumers aware of live streaming products, thereby increasing live streaming sales revenue. Secondly, when the cross price elasticity coefficient is low, the influencer anchor live streaming mode is the optimal choice. The substitutability of products from live streaming channels to products from online retail channels is small, hiring influencer anchors for live sales can bring more income. Thirdly, when the cross price elasticity coefficient is high, the substitutability of products from live streaming channels to products from online retail channels is higher. Compared with the higher commission rate of influencer anchors, the live streaming cost of merchant staff anchors is lower, merchant combined with virtual anchors live streaming mode is the optimal choice.

(3) When the increase of market share in live streaming channels, the online consumer group on live streaming channels is gradually growing. At this time, introducing virtual anchors can increase the exposure of products on live streaming platforms, attract more consumers to understand and purchase products, and improve the profits of manufacturers. Secondly, when the market share of live streaming channels is relatively small, the influencer anchor live streaming mode is the optimal choice, hiring influencer anchors for live streaming sales is more likely to attract consumers to purchase products and obtain more sales revenue. Thirdly, when the market share of live streaming channels is relatively large, based on the live sales level of merchant staff anchors and the advantage of continuous live streaming of virtual anchors, and the live streaming cost of merchant staff anchors is lower, the merchant combined with virtual anchors live streaming mode is the optimal choice.

(4) There is a positive relationship between manufacturers' profits and the consumer sensitivity to live streaming e-commerce, when the consumer sensitivity to live streaming e-commerce increases, introducing virtual anchors can leverage the novelty and uninterrupted live streaming of virtual anchors to enable manufacturers to obtain more profits. Secondly, when the consumer sensitivity to live streaming e-commerce is small, the introduction of virtual anchors has little effect and the live streaming cost of merchant staff anchors is lower. Therefore, the merchant live streaming mode is the optimal choice. Thirdly, when the consumer sensitivity to live streaming e-commerce is large, the influencer anchor combined with virtual anchors live streaming mode is the optimal live streaming mode. With the personal influence of influencer anchors and the advantage of continuous live streaming by virtual anchors, more consumers' attention and purchases can be attracted through live streaming channels.

Based on this study, the following management insights are obtained:

(1) Manufacturers should develop reasonable live streaming prices to increase product sales in live channels with reasonable price advantages. When the substitutability of products from live streaming channels to products from online retail channels is low, manufacturers can hire influencer anchors for product publicity, so as to increase product awareness and sales. When the live streaming products has a certain competitive advantage, the introduction of virtual anchors to increase the exposure of the product in the live platform, with the help of the price advantage and exposure, to attract more consumers to pay attention to and buy products.

(2) Manufacturers in order to attract more consumers to buy products from the live channel, can publicize the live product through short video platforms such as Jitterbug, Shutterbug, etc. to expand the influence of the product in the live room, and indirectly enhance

the consumer's willingness to buy the product from the live channel, combined with their own economic situation to introduce the virtual anchor, in the live platform to attract more consumers to pay attention to and buy.

(3) When the sensitivity of consumers to the products sold on live streaming is not yet very high, it is not suitable to introduce virtual anchors. Manufacturers should make efforts from the aspects of live streaming sales capacity enhancement, live streaming product quality assurance, after-sales service, etc., to continuously increase the consumers' favorability of live streaming products, and cultivate the word of mouth on the live streaming platform first, and then gradually introduce the virtual anchors afterwards, so as to increase the profit income.

## Author contributions

**Conceptualization:** Shizhen Bai, Man Jiang.

**Data curation:** Xiujin Gu.

**Formal analysis:** Xiujin Gu, Jinjin Zheng, Wenya Wu, Man Jiang, Ning Xue.

**Funding acquisition:** Shizhen Bai.

**Investigation:** Xiujin Gu.

**Methodology:** Xiujin Gu, Wenya Wu.

**Project administration:** Shizhen Bai.

**Resources:** Shizhen Bai.

**Software:** Xiujin Gu, Jinjin Zheng.

**Supervision:** Na Xu, Wenya Wu.

**Validation:** Jinjin Zheng, Wenya Wu.

**Visualization:** Xiujin Gu, Jinjin Zheng, Ning Xue.

**Writing – original draft:** Xiujin Gu.

**Writing – review & editing:** Na Xu, Wenya Wu.

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
