## [Decision Letter · Decision Letter 0]

14 Aug 2024

PONE-D-24-29296

A selection strategy for manufacturer's live streaming mode considering the characteristics of the live streaming anchor in the context of virtual anchor supplementation

PLOS ONE

Dear Dr. Wu,

Thank you for submitting your manuscript to PLOS ONE. After careful consideration, we feel that it has merit but does not fully meet PLOS ONE’s publication criteria as it currently stands. Therefore, we invite you to submit a revised version of the manuscript that addresses the points raised during the review process.

We look forward to receiving your revised manuscript.

Kind regards,

Vincenzo Basile, PhD

Academic Editor

PLOS ONE

   "Social Science Foundation Program of Heilongjiang province (Grant/Award Number 22GLD356)"

Reviewers' comments:

Reviewer's Responses to Questions

**Comments to the Author**

1. Is the manuscript technically sound, and do the data support the conclusions?

Reviewer #1: Partly

Reviewer #2: Yes

2. Has the statistical analysis been performed appropriately and rigorously? 

Reviewer #1: N/A

Reviewer #2: N/A

3. Have the authors made all data underlying the findings in their manuscript fully available?

Reviewer #1: No

Reviewer #2: Yes

4. Is the manuscript presented in an intelligible fashion and written in standard English?

Reviewer #1: No

Reviewer #2: Yes

5. Review Comments to the Author

Reviewer #1: After carefully reading through the manuscript, I feel that the topic is interesting and the workload is very rich. However, beyond that, I also noticed that there are many problems in this paper, such as a lack of sufficient analysis, weak contribution to real business operations, etc. My detailed comments are as follows and I hope they are helpful.

1、 The title of the paper is not concise enough.

2、 In the introduction section, the motivation of this paper should be explained clearly.

3、 The author needs to further explain why the e-retailer is the Stackelberg leader rather than the influencer anchor or the manufacturer?

4、 I don't quite understand the situation where the influencer anchor live streaming selling combined with the virtual anchor mode, are there any real-life cases to support this?

5、 I don't quite understand why the author defines the number of anchors t and why t<30?

6、 In the analysis section, the authors' explanation of the proposition focuses more on explaining the results and lacks exploration of the underlying reasons. The explanation of propositions is more about explaining why such results are produced, which factors or effects play an important role in it, so as to provide a basis for subsequent management insights.

7、 The innovation of this paper is not clear. There are many existing literature studies on the live streaming e-commerce. So, the specific contributions and innovations of this article in terms of research perspective, research conclusions, etc. need to be clearly explained.

8、 The research conclusions of this article are relatively intuitive, and no innovative conclusions have been obtained from the study.

9、 The language is not smooth enough, it is recommended to polish the overall language of the paper.

Reviewer #2: 1. In the Abstract, add a clear statement of the paper's unique contribution, e.g., "This study is the first to incorporate virtual anchors into live streaming mode selection strategies."

2. Introduction:

(1) In lines 41-42, include global examples beyond Taobao Live, Facebook Live, and Amazon Live. Consider mentioning platforms like Instagram Live Shopping or Twitter's "Shop Module".

(2) After line 57, add a paragraph summarizing key literature on live streaming e-commerce strategies to establish the research context.

(3) On line 77, provide a brief explanation of how the Stackelberg game method applies to this study.

3. Structure:

(1) After the Introduction, add the following sections in order: Literature Review, Methodology, Results, Discussion, Conclusion.

(2) At the end of the Introduction, add a paragraph outlining the paper's structure.

4. Methodology:

(1) After line 94, clearly define each of the four live streaming modes and explain how they were derived.

(2) Provide full mathematical formulations for the Stackelberg game model used in the study.

5. Results:

(1) Create a table summarizing the optimal profits for e-tailers and manufacturers under each of the four live streaming modes.

(2) Include graphs showing how key variables (e.g., price competition intensity, consumer sensitivity) affect mode selection.

6. Discussion:

(1) After presenting the results, add a section comparing your findings with those of previous studies, such as the work mentioned in lines 64-67 about Gree Electric Chairman Dong Mingzhu's live streaming success.

7. Limitations and Future Research:

Add a new section before the Conclusion discussing at least three limitations of the current study and suggesting corresponding future research directions.

8. Language:

(1) Throughout the paper, ensure consistent use of terms. For example, decide between "live streaming selling" and "live e-commerce" and use the chosen term consistently.

(2) In lines 20-22, revise the sentence structure for clarity: "This paper, in the context of virtual anchors supplementing human anchors in live streaming e-commerce, combines the characteristics of live streaming anchors and the number of human anchor live streams to construct a model of four live streaming modes."

9. Practical Implications:

In the Conclusion, add a paragraph specifically addressing how manufacturers can use the study's findings to select the most appropriate live streaming mode based on their specific circumstances (e.g., market share, available resources).

6. PLOS authors have the option to publish the peer review history of their article (what does this mean? ). If published, this will include your full peer review and any attached files.

**Do you want your identity to be public for this peer review?** For information about this choice, including consent withdrawal, please see our Privacy Policy .

Reviewer #1: No

Reviewer #2: No

---

## [Author Response · Author response to Decision Letter 1]

16 Oct 2024

Dear Editor and Reviewers,

Thanks very much for taking your time to review this manuscript. We really appreciate all your comments and suggestions. In accordance with the instructions in your letter, we have uploaded the revised version of the manuscript with all the changes highlighted by using the track changes mode in MS Word. Appended to this letter is our itemized response to the comments raised by two reviewers. The comments from reviewers are reproduced in italic type and our responses are given directly afterward in a different color (red). Thanks again!

Reviewer #1

Q1: The title of the paper is not concise enough.

Answer1: Thank you for your suggestions, we have simplified the title of the paper and now modify it to:

Strategies for pricing and mode selection for manufacturer live streaming with virtual anchor supplementation.

Q2: In the introduction section, the motivation of this paper should be explained clearly.

Answer2: We thank the reviewer for pointing this out. Our explanation of the motivation for writing the paper in the introduction section is supplemented as follows (Lines 56-62 on page 3):

In the context of the virtual anchors as a supplement to human anchors in live streaming e-commerce, the merchant combined with virtual anchors live streaming mode, influencer anchor combined with virtual anchors live streaming mode, while playing the advantages of human anchors and virtual anchors, but the virtual anchor live interaction ability is limited, and need to invest a certain cost, after the introduction of virtual anchors, how the price of live streaming should be set, and how to choose between different live streaming modes remains to be studied. (Lines 56-62 on page 3).

Q3: The author needs to further explain why the e-retailer is the Stackelberg leader rather than the influencer anchor or the manufacturer?

Answer3: Thank you for your suggestions, Our explanation regarding e-tailer as the leader of Stackelberg is as follows(Lines 239-250 on page 12):

When a manufacturer establishes a live streaming channel, it will seize the market share of the existing e-tailers, and there is a certain competitive relationship between the manufacturer and the e-tailers. Compared with the e-tailers, the manufacturer is in a low competitive position due to the fact that the online consumer group it has accumulated and cultivated is not yet in scale. The influencer anchors are an option for manufacturers, and they are in a short-term employment relationship, leaving out the participation of influencer anchors in gaming for the time being. Therefore, this paper assumes that the Stackelberg game is played between the manufacturer and the e-tailer, with the e-tailer as the dominant player and the manufacturer as the follower, the e-tailer first determines the price of the e-tailing channel, and then the manufacturer sets the price of the live streaming channel, and then the consumer finally decides which sales channel to buy the product from. (Lines 239-250 on page 12).

Q4: I don't quite understand the situation where the influencer anchor live streaming selling combined with the virtual anchor mode, are there any real-life cases to support this?

Answer4: Thank you for your suggestions, Our supplementary explanation on the influencer anchor combined with virtual anchors for internet celebrities is as follows(Lines 50-55 on page 3)：

For example, Kazilan, Perrier, perfect diary and other well-known beauty brands, choose merchants staff anchor or Viya, Li Jiaqi and other influencer anchor live sales at the same time, after 12 midnight, will also use “big eyes Kaka” and other virtual anchor instead of the human anchor to continue to sell the product, to fill the live streaming time gap, this creates a new space for revenue growth. (Lines 50-55 on page 3).

Q5: I don't quite understand why the author defines the number of anchors t and why t<30?

Answer5: Thank you for your suggestions, Our supplementary explanation on the influencer anchor combined with virtual anchors for internet celebrities is as follows：

Thank you for your suggestions, the main purpose of setting the number of live streaming of human anchors in the model is to indicate that virtual anchors have the advantage of not being limited by time and continuously broadcasting to increase product exposure compared to human streamers. After considering that the Number of live streaming of human anchors cannot effectively demonstrate the impact of continuous streaming by virtual anchors on increasing product exposure，and the continuous live streaming of virtual anchors, due to their novelty[1,2] and the exposure effect, will to some extent affect consumers' willingness to purchase[3]. so replace the Number of live streaming of human anchors with Level of investment in continuous live streaming for virtual anchors, Level of investment in continuous live streaming for virtual anchors indicates the advantage of virtual anchors not being limited by time and increasing product exposure through continuous live streaming. The specific changes to the variables and model related parts are described below.

The demand functions for the merchant combined with virtual anchors live streaming mode, and the influencer anchor combined with virtual anchors live streaming mode, have been adjusted accordingly (Lines 295-306 on page 15).

The profit function for the merchant combined with virtual anchors live streaming mode, and the influencer anchor combined with virtual anchors live streaming mode, have been adjusted accordingly (Lines 371-373 on page 20; Lines 402-404 on page 22).

References

1. Gao JY, Zhao XJ, Zhai MF, Zhang D, Li G. AI or Human? The Effect of Streamer Types on Consumer Purchase Intention in Live Streaming. International Journal of Human-Computer Interaction. 2024.

2. Zhang YX, Wang XY, Zhao X. Supervising or assisting? The influence of virtual anchor driven by AI-human collaboration on customer engagement in live streaming e-commerce. Electronic Commerce Research. 2023.

3. Xie K , Lee Y J .Social Media and Brand Purchase: Quantifying the Effects of Exposures to Earned and Owned Social Media Activities in a Two-Stage Decision Making Model.Journal of Management Information Systems. 2015; 32(2):204-238.

Q6: In the analysis section, the authors' explanation of the proposition focuses more on explaining the results and lacks exploration of the underlying reasons. The explanation of propositions is more about explaining why such results are produced, which factors or effects play an important role in it, so as to provide a basis for subsequent management insights.

Answer6: Thank you for your suggestions, We provide additional explanations on the innovative points of our research in this article，Here are some of the contents（Lines 514-530 on page 30；Lines 531-542 on page 31）.with all changes reflected in the main text（Lines 550-552 on page 31；Lines 553-573 on page 32；Lines 574 on page 33；Lines 583-595 on page 33；Lines 596-610 on page 34）.

As shown in Fig3, the manufacturer's profit is directly proportional to the cross price elasticity coefficient. When the cross price elasticity coefficient changes, the profit changes of the four live streaming modes are analyzed and expressed as follows. Firstly, as the cross price elasticity coefficient increases, the profit of the merchant combined with virtual anchors live streaming mode (the influencer anchor combined with virtual anchors live streaming mode) is higher than that of the merchant live streaming mode (the influencer anchor live streaming mode). When the cross price elasticity coefficient increases, the change in demand for products in online retail channels is more easily affected by the price of live streaming products. The higher the substitutability of products in live streaming channels to products in online retail channels, the more manufacturers can supplement live streaming sales with virtual anchors, increase live streaming exposure, and make more consumers aware of live streaming products, thereby increasing live streaming sales revenue. Secondly, when the cross price elasticity coefficient is low, the influencer anchor live streaming mode is the optimal choice. When the cross price elasticity coefficient is low, the substitutability of products from live streaming channels to products from online retail channels is small, and products from live streaming channels do not have product advantages. At this time, merchant staff anchors do not have enough attractiveness to consumers, while influencer anchors can attract more consumer attention and purchase products due to their personal influence. Hiring influencer anchors for live sales can bring more income. Thirdly, when the cross price elasticity coefficient is high, the merchant combined with virtual anchors live streaming mode is the optimal choice. When the cross price elasticity coefficient is large, the substitutability of products from live streaming channels to products from online retail channels is higher, and products from live streaming channels have a certain competitive advantage. Compared with the higher commission rate of influencer anchors, the live streaming cost of merchant staff anchors is lower, and manufacturers gain more profits. Therefore, when the cross price elasticity coefficient is large, merchant combined with virtual anchors live streaming mode is the optimal choice.（Lines 514-530 on page 30；Lines 531-542 on page 31）.

Q7: The innovation of this paper is not clear. There are many existing literature studies on the live streaming e-commerce. So, the specific contributions and innovations of this article in terms of research perspective, research conclusions, etc. need to be clearly explained.

Answer7: We thank the reviewer for pointing this out. The specific supplementary explanation for the innovative points of this study is as follows (Lines 197-208 on page 10):

Different from previous studies, this paper, in the context of virtual anchors as a supplement to human anchors in live streaming e-commerce, considers the introduction of virtual anchors from merchant live streaming mode, influencer anchor live streaming mode, respectively, and constructs four different live streaming modes, which enriches the live streaming mode model. Secondly, for the first time, virtual anchors are included in the live streaming pricing and mode selection decisions, and the optimal live streaming channel pricing of manufacturers is studied by synthesizing live streaming anchor characteristics and live streaming costs. Finally, based on the impact of parameters such as cross price elasticity coefficient, market share of live streaming channels and consumer sensitivity to live streaming e-commerce on manufacturers' profits, it provides a reference for manufacturer's live streaming mode selection decisions. (Lines 197-208 on page 10).

Q8: The research conclusions of this article are relatively intuitive, and no innovative conclusions have been obtained from the study.

Answer8: Thank you for your suggestions, We have further supplemented the lack of innovation in our research conclusions，Here are some of the contents（Lines 665-677 on page 37）.with all changes reflected in the main text（Lines 650-658 on page 36；Lines 659-679 on page 37; Lines 680-700 on page 37）:

When the cross price elasticity coefficient increases, the higher the substitutability of products in live streaming channels to products in online retail channels, the manufacturers can supplement live streaming sales with virtual anchors, increase live streaming exposure, and make more consumers aware of live streaming products, thereby increasing live streaming sales revenue. Secondly, when the cross price elasticity coefficient is low, the influencer anchor live streaming mode is the optimal choice. The substitutability of products from live streaming channels to products from online retail channels is small, hiring influencer anchors for live sales can bring more income. Thirdly, when the cross price elasticity coefficient is high, the substitutability of products from live streaming channels to products from online retail channels is higher. Compared with the higher commission rate of influencer anchors, the live streaming cost of merchant staff anchors is lower, merchant combined with virtual anchors live streaming mode is the optimal choice. （Lines 665-677 on page 37）.

Q9: The language is not smooth enough, it is recommended to polish the overall language of the paper.

Answer9: Thank you for your suggestions, we have revised and polished the overall language of the paper.

Reviewer #2

Q1: In the Abstract, add a clear statement of the paper's unique contribution, e.g., "This study is the first to incorporate virtual anchors into live streaming mode selection strategies."

Answer1: We thank the reviewer for pointing this out. Our explanation of the unique contribution of this article in the abstract is as follows（Lines 2-6 on page 1）:

In the context of virtual anchors as a supplement to human anchors in live streaming e-commerce. In this paper, for the first time, virtual anchors are included in the live streaming pricing and mode selection strategy, combined with the characteristics of the live streaming anchor and the cost of live streaming, constructed a model of the four live streaming modes, and used the Stackelberg game method to study.（Lines 2-6 on page 1）.

Q2.1: In lines 41-42, include global examples beyond Taobao Live, Facebook Live, and Amazon Live. Consider mentioning platforms like Instagram Live Shopping or Twitter's "Shop Module".

Answer2.1: Thank you for your suggestions, we have supplemented the examples of Instagram Live Shopping or Twitter's "shopping module" in the introduction, We have supplemented the examples of Instagram Live Shopping or Twitter's "shopping module" in the introduction, the specific supplementary content is as follows （Lines 24-28 on page 2）:

Live streaming is becoming increasingly popular globally [1], with both merchants and consumers selling and buying products on platforms such as Instagram Live Shopping, Twitter's e-commerce module, Taobao Live, Facebook Live, and Amazon Live in their daily lives [2], which has given rise to a new form of business called live streaming commerce [3].（Lines 24-28 on page 2）.

Q2.2: After line 57, add a paragraph summarizing key literature on live streaming e-commerce strategies to establish the research context.

Answer2.2: We thank the reviewer for pointing this out. A summary and explanation of key literature on live streaming e-commerce strategies, as follows (Lines 37-42 on page 2):

When merchants carry out live sales services, they need to determine the live streaming price and live streaming mode, current scholars from the live streaming anchor type, live streaming anchor characteristics, consumer preferences, consumer behavior and other perspectives, with the help of the Stackelberg game approach to research, for the manufacturer's live streaming pricing and mode selection to provide the relevant decision support. (Lines 37-42 on page 2).

Q2.3: On line 77, provide a brief explanation of how the Stackelberg game method applies to this study.

Answer2.3: We thank the reviewer for pointing this out. The supplementary explanation on the application of Stackelberg game method in this study is as follows (Lines 70-74 on page 4):

In addition, the live streaming channel, as a new sales channel, will seize the market share of the existing e-tailers, and there is a certain competitive relationship between manufacturers and e-tailers, so the Stackelberg game method is applied to study the live streaming pricing and mode selection problem of manufacturers. (Lines 70-74 on page 4).

Q3.1: After the Introduction, add the following sections in order: Literature Review, Methodology, Results, Discussion, Conclusion.

Answer3.1: Thank you for your suggestions. We have adjusted the structure order of this article according to the chapters of Introduction, Literature Review, Methodology, Results, Discussion, and Conclusion, and improved the content arrangement of the Methodology and Results chapters. At this point, the metho

---

## [Decision Letter · Decision Letter 1]

20 Dec 2024

PONE-D-24-29296R1Strategies for pricing and mode selection for manufacturer live streaming with virtual anchor supplementationPLOS ONE

Dear Dr. Wu,

Thank you for submitting your manuscript to PLOS ONE. After careful consideration, we feel that it has merit but does not fully meet PLOS ONE’s publication criteria as it currently stands. Therefore, we invite you to submit a revised version of the manuscript that addresses the points raised during the review process.

We look forward to receiving your revised manuscript.

Kind regards,

Vincenzo Basile, PhD

Academic Editor

PLOS ONE

Journal Requirements:

Reviewers' comments:

Reviewer's Responses to Questions

**Comments to the Author**

1. If the authors have adequately addressed your comments raised in a previous round of review and you feel that this manuscript is now acceptable for publication, you may indicate that here to bypass the “Comments to the Author” section, enter your conflict of interest statement in the “Confidential to Editor” section, and submit your "Accept" recommendation.

Reviewer #1: (No Response)

Reviewer #3: All comments have been addressed

2. Is the manuscript technically sound, and do the data support the conclusions?

Reviewer #1: (No Response)

Reviewer #3: Yes

3. Has the statistical analysis been performed appropriately and rigorously? 

Reviewer #1: (No Response)

Reviewer #3: Yes

4. Have the authors made all data underlying the findings in their manuscript fully available?

Reviewer #1: (No Response)

Reviewer #3: Yes

5. Is the manuscript presented in an intelligible fashion and written in standard English?

Reviewer #1: (No Response)

Reviewer #3: Yes

6. Review Comments to the Author

Reviewer #1: After careful reading, I found that the author has made some modifications to the paper based on the opinions, but I believe that the understanding and revision of some issues cannot reach a satisfactory level. Therefore, I need to make a minor revision decision and hope that the author can take it seriously.

1、 The author's modifications to the title did not reach a satisfactory level, and even the revised title became ambiguous.

2、 The author's answer to question 2 is too simple and does not present the motivation of this study logically and clearly. I suggest the author to reorganize the entire introduction section.

3、 I think the author's explanation of question 3 is unreasonable or the author did not answer this question. I suggest the author to explain it from both theoretical and practical perspectives.

Reviewer #3: Referee Report on “Strategies for pricing and mode selection for manufacturer live streaming with virtual anchor supplementation”

Manuscript ID: PONE-D-24-29296R1

Summary and Comment:

This paper examines the pricing and mode selection strategies for manufacturers utilizing live streaming with virtual anchor supplementation. It constructs a model encompassing four live streaming modes: merchant live streaming, influencer anchor live streaming, merchant live streaming combined with virtual anchors, and influencer anchor live streaming combined with virtual anchors. The study employs the Stackelberg game approach to analyze how factors such as cross-price elasticity, market share of live streaming channels, and consumer sensitivity to live streaming e-commerce influence the optimal pricing and mode selection for manufacturers.

The topic is both interesting and significant to this field. The responses to the reviewers' questions are satisfactory and reflect the reviewers' concerns. I did not identify any substantial errors or issues in this study. Therefore, the paper can be accepted in its current form.

7. PLOS authors have the option to publish the peer review history of their article (what does this mean? ). If published, this will include your full peer review and any attached files.

**Do you want your identity to be public for this peer review?** For information about this choice, including consent withdrawal, please see our Privacy Policy .

Reviewer #1: No

Reviewer #3: No

---

## [Author Response · Author response to Decision Letter 2]

7 Feb 2025

Dear Editor and Reviewers,

Thanks very much for taking your time to review this manuscript. We really appreciate all your comments and suggestions. In accordance with the instructions in your letter, we have uploaded the revised version of the manuscript with all the changes highlighted by using the track changes mode in MS Word. Appended to this letter is our itemized response to the comments raised by the reviewer. The comments from reviewers are reproduced in italic type and our responses are given directly afterward in a different color (red). Thanks again!

Reviewer #1

Q1: The author's modifications to the title did not reach a satisfactory level, and even the revised title became ambiguous.

Answer1: Thank you for your suggestions, we have simplified the title of the paper and now modify it to:

Live streaming mode selection strategy under the background of virtual anchor supplementation.

Q2: The author's answer to question 2 is too simple and does not present the motivation of this study logically and clearly. I suggest the author to reorganize the entire introduction section.

Answer2: We thank the reviewer for pointing this out. We have reorganized the content of the entire introduction section, and the relevant content is as follows (Lines 45-64 on page 3; Lines 65-85 on page 4; Lines 86-92 on page 5):

Live streaming is becoming increasingly popular worldwide [1]. In daily life, both businesses and consumers can sell and purchase products on platforms such as Instagram Live Shopping, Twitter's e-commerce module, Taobao Live, Facebook Live, and Amazon Live [2]. With the development of live streaming e-commerce [3], new live streaming modes are constantly emerging. Businesses can choose suitable live streaming modes based on factors such as live streaming anchor type, live streaming anchor characteristics, and consumer preferences [4,5,6,7], such as merchant live streaming mode, influencer anchor live streaming mode, etc. With the development of artificial intelligence technology, using virtual anchors as a supplement to human anchors has become a new hybrid live streaming mode. For example, well-known beauty brands such as Carslan, L'Oreal, and Perfect Diary choose merchant staff anchors or influencer anchors such as Viya and Li Jiaqi for live streaming sales. After midnight, they also use virtual anchors such as "Big Eye Kaka", "Ou Xiaomi", and "Stella" to replace human anchors and continue selling products.

The mixed live streaming mode of human anchors and virtual anchors can use virtual anchors to fill the time gap when human anchors cannot live streaming sales. However, the construction and application of virtual anchors require a certain cost investment. At present, the low-end version of virtual anchors on the market is priced at tens of thousands of yuan. Merchants can achieve "7 * 24" hour live streaming sales of virtual anchors by purchasing anchor software and operation and maintenance services. The high-end version of virtual anchors is mostly personalized customization, with different levels of pricing based on the difficulty of customization, with an average price of around 200000 yuan. Due to the poor live streaming interaction ability of virtual anchors, most live streaming rooms choose to use a combination of "human anchors and virtual anchors" for live streaming sales, manufacturers need to pay for the construction and application costs of virtual anchors in addition to the cost of human anchors. Therefore, based on cost and profit considerations, whether manufacturers choose traditional live streaming modes such as merchant live streaming mode and influencer anchor live streaming mode, or choose to invest in and apply virtual anchors on the basis of the traditional live streaming mode to obtain additional income, is a question worth studying.

Based on this, in the context of virtual anchors as a supplement to human anchors in live streaming e-commerce. In this paper, the first time to include virtual anchors in the live streaming mode selection strategy, combined with the characteristics of the live streaming anchor and the cost of live streaming, four live streaming modes are constructed: merchant live streaming mode, influencer anchor live streaming mode, merchant combined with virtual anchors live streaming mode, and influencer anchor combined with virtual anchors live streaming mode. Explore the optimization problem of manufacturer's live streaming mode selection decision. The problems to be solved are as follows:

(1) What are the optimal live streaming channel prices and manufacturers' optimal profits under different live streaming modes?

(2) What impact do parameters such as cross price elasticity coefficient, market share of live streaming channels and consumer sensitivity to live streaming e-commerce have on live streaming channel prices?

(3) What is the manufacturer's live streaming mode selection strategy considering the impact of parameters such as cross price elasticity coefficient, market share of live streaming channels and consumer sensitivity to live streaming e-commerce?

Q3: I think the author's explanation of question 3 is unreasonable or the author did not answer this question. I suggest the author to explain it from both theoretical and practical perspectives.

Answer3: Thank you for your suggestions. We have made corrections to the issue of viewing e-tailers as Stackelberg leaders, as follows (Lines 236-254 on page 12):

In the context of virtual anchors as a supplement to human anchors in live streaming e-commerce, when manufacturers choose to expand their sales channels through live streaming sales, there are four modes: merchant live streaming mode, influencer anchor live streaming mode, merchant combined with virtual anchors live streaming mode, and influencer anchor combined with virtual anchors live streaming mode. In the merchant live streaming mode and the merchant combined with virtual anchors live streaming mode, manufacturers play a dominant role in the game with e-tailers based on their strong financial strength and long-term product development [35,36,37]. Therefore, when manufacturers establish live streaming channels to sell products, they will prioritize determining the live streaming channel price, and e-tailers will determine the e-tailing channel price based on the manufacturer's live streaming pricing; In the influencer anchor live streaming mode and the influencer anchor combined with virtual anchors live streaming mode, manufacturers hire influencer anchors for live streaming sales, such as well-known influencer anchors with a large number of fans such as Li Jiaqi, Viya, Dong Yuhui, etc. Due to their influence, they have certain pricing power in live streaming sales [38,39]. Therefore, when manufacturers establish live streaming channels to sell products, influencer anchors will first determine the live streaming channel price, and e-tailers will determine the e-tailing channel price based on the live streaming pricing of influencer anchors.

In addition, based on the latest game sequence, we have also made relevant modifications to the model results and simulation analysis. Among them, the model results have changed slightly, while the content of simulation analysis has not changed much. Some modifications are as follows (Lines 316-331 on page 16; Lines 332-345 on page 17), and all modifications have been reflected in the main text (Lines 236-254 on page 12; Lines 316-331 on page 16; Lines 332-345 on page 17; Lines 348-350 on page 17; Lines 351-367 on page 18; Lines 368-381 on page 19; Lines 384-399 on page 20; Lines 400-412 on page 21; Lines 416-432 on page 22; Lines 433-446 on page 23; Lines 447-455 on page 24; Lines 456-473 on page 25; Lines 474-493 on page 26; Lines 494-505 on page 27).

---

## [Decision Letter · Decision Letter 2]

25 Feb 2025

PONE-D-24-29296R2Live streaming mode selection strategy under the background of virtual anchor supplementationPLOS ONE

Dear Dr. Wu,

Thank you for submitting your manuscript to PLOS ONE. After careful consideration, we feel that it has merit but does not fully meet PLOS ONE’s publication criteria as it currently stands. Therefore, we invite you to submit a revised version of the manuscript that addresses the points raised during the review process.

**ACADEMIC EDITOR: ** The reviewers have shown a positive inclination toward your work, with Reviewer #1 confirming that all concerns have been adequately addressed and recommending the paper for publication in its present form. However, Reviewer #3 has suggested minor revisions before final acceptance. Specifically, the reviewer has requested improvements in figure clarity, as the current resolution in the PDF is unclear, and an explanation of the role of consumer trust and engagement with virtual anchors in influencing purchasing behavior during live-streaming events, along with insights into how these factors can be quantified within your model. We kindly request you to revise the manuscript accordingly and submit an updated version along with a point-by-point response addressing these concerns. 

We look forward to receiving your revised manuscript.

Kind regards,

Jitendra Yadav, Ph.D.

Academic Editor

PLOS ONE

Journal Requirements:

Reviewers' comments:

Reviewer's Responses to Questions

**Comments to the Author**

1. If the authors have adequately addressed your comments raised in a previous round of review and you feel that this manuscript is now acceptable for publication, you may indicate that here to bypass the “Comments to the Author” section, enter your conflict of interest statement in the “Confidential to Editor” section, and submit your "Accept" recommendation.

Reviewer #1: (No Response)

Reviewer #3: All comments have been addressed

2. Is the manuscript technically sound, and do the data support the conclusions?

Reviewer #1: (No Response)

Reviewer #3: Yes

3. Has the statistical analysis been performed appropriately and rigorously? 

Reviewer #1: (No Response)

Reviewer #3: Yes

4. Have the authors made all data underlying the findings in their manuscript fully available?

Reviewer #1: (No Response)

Reviewer #3: No

5. Is the manuscript presented in an intelligible fashion and written in standard English?

Reviewer #1: (No Response)

Reviewer #3: Yes

6. Review Comments to the Author

Reviewer #1: All my concerns have been addressed. The paper is in good shape now and is ready for publication. I am happy to recommend its publication in its present form.

Reviewer #3: Referee Report on “Live streaming mode selection strategy under the background of virtual anchor supplementation”

Manuscript ID: PONE-D-24-29296R2

Summary and Comment:

The manuscript offers a fresh perspective on enhancing live-streaming e-commerce through the use of virtual anchors. It develops a model that includes four distinct live-streaming approaches and utilizes Stackelberg game theory to evaluate optimal pricing and mode selection tactics. This study enriches existing literature by embedding virtual anchors within the decision-making processes of live commerce, taking into account factors like market share, consumer responsiveness, and cost dynamics.

The manuscript provides a robust theoretical framework for selecting live-streaming modes with the inclusion of virtual anchors. To enhance its empirical validation, the study should incorporate insights into consumer behavior and address ethical concerns, which would significantly increase its impact and applicability.

The topic is both interesting and significant to this field. The responses to the reviewers' questions are satisfactory and adequately address their concerns. I did not identify any substantial errors or issues in this study. Therefore, the paper can be accepted with minor revisions.

Questions for the Authors:

1. The figures used in the study require refinement, as they are currently unclear in the PDF file. Please enhance the resolution of the images.

2. What role do consumer trust and engagement with virtual anchors play in influencing purchasing behavior during live-streaming events, and how can these factors be quantified within the model? Please explain.

7. PLOS authors have the option to publish the peer review history of their article (what does this mean? ). If published, this will include your full peer review and any attached files.

**Do you want your identity to be public for this peer review?** For information about this choice, including consent withdrawal, please see our Privacy Policy .

Reviewer #1: No

Reviewer #3: No

---

## [Author Response · Author response to Decision Letter 3]

27 Feb 2025

Dear Editor and Reviewers,

Thanks very much for taking your time to review this manuscript. We really appreciate all your comments and suggestions. In accordance with the instructions in your letter, we have uploaded the revised version of the manuscript with all the changes highlighted by using the track changes mode in MS Word. Appended to this letter is our itemized response to the comments raised by the reviewer. The comments from reviewers are reproduced in italic type and our responses are given directly afterward in a different color (red). Thanks again!

Reviewer #3

Q1: The figures used in the study require refinement, as they are currently unclear in the PDF file. Please enhance the resolution of the images.

Answer1: Thank you for your suggestions, we have enhanced the resolution of the images.

Q2: What role do consumer trust and engagement with virtual anchors play in influencing purchasing behavior during live-streaming events, and how can these factors be quantified within the model? Please explain.

Answer2: We thank the reviewer for pointing this out. Our supplementary explanation of this issue is as follows (Lines 236-244 on page 12):

Virtual anchors are a new type of live streaming anchor [22], and consumers' trust in virtual anchors can enhance their willingness to purchase [35]. In addition, consumers also have trust issues with the live streaming sales performance of ordinary live streaming anchors, influencer anchors, and other live streaming anchors [28]. Therefore, for ease of analysis, this article does not separately consider consumers' trust in virtual anchors as an influencing factor, but uniformly regards consumers' trust and sensitivity to the characteristics of live streaming anchors (trustworthiness, live streaming interaction ability, etc.) as consumers' sensitivity to live streaming e-commerce [5,28], and uses to represent it.

References:

5. Pan R, Feng J, Zhao ZL. Fly with the wings of live-stream selling-Channel strategies with/without switching demand. Production and Operations Management. 2022;31(9):3387-99.

22. Gao W, Jiang N, Guo QQ. How do virtual streamers affect purchase intention in the live streaming context? A presence perspective. Journal of Retailing and Consumer Services. 2023;73.

28. Ladhari R, Massa E, Skandrani H. YouTube vloggers' popularity and influence: The roles of homophily, emotional attachment, and expertise. Journal of Retailing and Consumer Services. 2020;54.

35. Yuan, L., Dennis, A.R. Acting like humans? Anthropomorphism and consumer's willingness to pay in electronic commerce. Journal of Management Information Systems. 2019; 36: 450-477.

---

## [Editor Report · Decision Letter 3]

9 Mar 2025

Live streaming mode selection strategy under the background of virtual anchor supplementation

PONE-D-24-29296R3

Dear Dr. Wu,

We’re pleased to inform you that your manuscript has been judged scientifically suitable for publication and will be formally accepted for publication once it meets all outstanding technical requirements.

Kind regards,

Jitendra Yadav, Ph.D.

Academic Editor

PLOS ONE
---

## [Editor Report · Acceptance letter]

PONE-D-24-29296R3

PLOS ONE

Dear Dr. Wu,

I'm pleased to inform you that your manuscript has been deemed suitable for publication in PLOS ONE. Congratulations! Your manuscript is now being handed over to our production team.

Kind regards,

on behalf of

Dr. Jitendra Yadav

Academic Editor

PLOS ONE